# RNF20 links the DNA damage response and metabolic rewiring in lung cancer through HIF1α

Hao Liu[1,21], Yongqin Tang[1,21], Anshu Singh[1,2,21], Joaquim Vong [2], Julio Cordero[1], Arthur Mathes[1], Rui Gao [1,2], Yanhan Jia[1,2], Boyan K. Garvalov [3], Till Acker [4], Gernot Poschet [5], Rüdiger Hell [5], Marc A. Schneider [6,7], Joerg Heineke [8,9], Thomas Wieland [9,10], Guillermo Barreto [11], Adelheid Cerwenka [12,13], Michael Potente [2,14,15], Sofia-Iris Bibli[16,17], Rajkumar Savai [18,19] & Gergana Dobreva [1,2,16,20] ✉

Defective DNA repair and metabolic rewiring are highly intertwined in promoting the development and progression of cancer. However, the molecular players at their interface remain poorly understood. Here we show that an RNF20-HIF1α axis links the DNA damage response and metabolic reprogramming in lung cancer. We demonstrate that RNF20, which catalyzes mono-ubiquitylation of histone H2B (H2Bub1), controls *Rbx1* expression and thereby the activity of the VHL ubiquitin ligase complex and HIF1α levels. Ablation of a single *Rnf20* allele significantly increases the incidence of lung tumors in mice. Mechanistically, *Rnf20* haploinsufficiency results in inadequate tumor suppression via the Rnf20-H2Bub1-p53 axis and induces DNA damage, cell growth, epithelial-mesenchymal transition (EMT), and metabolic rewiring through HIF1α-mediated RNA polymerase II promoter-proximal pause release, which is independent of H2Bub1. Importantly, decreased RNF20 levels correlate with increased expression of HIF1α and its target genes, suggesting HIF1α inhibition as a promising therapeutic approach for lung cancer patients with reduced RNF20 activity.

DNA repair and metabolic pathways undergo extensive changes during tumorigenesis, often resulting in genetic heterogeneity, rapid growth, and invasive behavior. Therefore, increasing effort is being invested in the development of strategies for targeting the DNA damage response[1–5] or cell metabolism[6,7] in cancer.

Acute DNA damage leads to transient cell-cycle arrest and DNA damage repair, or activates cell death or senescence[3,8–10]. The p53 and RB tumor-suppressor pathways play a central role in the protection from potentially damaging oncogenic stimuli, by controlling cell-cycle checkpoints, cell senescence, and death[11–13]. Therefore, inherited defects in the DNA-damage response lead to accumulation of genetic mutations, genomic instability, and a higher cancer risk[3].

On the other hand, rewiring of energy production networks in tumor cells supports rapid proliferation, continuous growth, survival, invasion, and metastasis. In cancer cells, oxidative phosphorylation is often inhibited and cells use glycolysis to generate energy, producing lactate as the end product[14–16]. Lactate promotes cancer cell survival, invasion, metastasis, and influences the tumor microenvironment[17]. Cancer cells upregulate the expression of most glycolytic enzymes, mainly through activation of two master inducers of glycolysis, c-Myc and HIF1α, and inadequate p53-mediated control[18,19]. c-Myc and HIF1α both bind directly to DNA regulatory regions of a large number of glycolytic genes, but while HIF1α is mainly active in hypoxia, c-Myc promotes the expression of its

glycolytic target genes also in normoxia[18,20]. Thus, a dynamic interplay between these master regulators continuously drives glycolysis to support rapid proliferation and accelerated anabolism of cancer cells[15,16,18,21].

Recent studies suggest that the DNA-damage response and metabolic rewiring are tightly interconnected as a part of the cellular response to genotoxic stress[22,23]. The molecular machinery at the interface of these two key processes responsible for tumorgenesis is poorly understood, but holds great promise in anti-cancer therapies.

Epigenetic mechanisms play a key role in the regulation not only of transcription but also of genomic stability[24,25]. It has been previously shown that monoubiquitination of H2B at DNA damage sites by the E3-ligase complex RNF20/RNF40[26] is required for recruitment of double-strand break (DSB) repair proteins and timely DSB

**Fig. 1 | Spontaneous lung tumor formation in *Rnf20* +/- mice. a** Histological analysis of representative lung sections of wild-type (*n* = 20) and *Rnf20* +/- (*n* = 32) mice at one year of age; tumors are indicated by arrows. Scale bars, 200 μm. 1, 2 – normal lung tissue; 3, 5 – lesions with adenocarcinoma (AD) characteristics; 4, 6 – lesions with small cell lung carcinoma (SCLC) characteristics. **b** High magnification images of lesions with AD characteristics (left) and with SCLC characteristics (right). Scale bars, 20 μm. **c** Immunohistochemical staining of lungs of *Rnf20* +/- mice for CGRP, Syp, and CD45 (top panels, 1, 2, 3); TTF1, CD45, and Ki67 (middle panels, 4, 5, 6). Scale bars, 200 μm. The bottom panels show immunostaining for TTF1, Napsin A, and p63 (bottom panels, 7, 8, 9). Scale bars, 100 μm. **d** Immunohistochemistry of lung tumor tissue samples from a microarray stained with an anti-RNF20 antibody. Scale bars, 100 μm. **e** Relative staining intensity (H-Score) for RNF20 in lung tumors of different types and grades. *n* = 70 for AD (*n* = 20 Grade 1, *n* = 30 Grade 2, *n* = 20 Grade 3); *n* = 22 for SCLC and *n* = 20 for normal lung (NL). **f, h** Normalized expression of *RNF20* in normal (*n* = 68) and AD (*n* = 524) (**f**) or *RNF40* (**h**) in normal (*n* = 59) and AD (*n* = 515) patients in TCGA datasets. FPKM, fragments per kilobase of exon model per million mapped fragments. **g, i** Overall survival (Kaplan-Meier plot) of lung AD patients[35] expressing high (*n* = 336) vs low (*n* = 336) levels of *RNF20* (probe 222683_at) (**g**) or high (*n* = 580) vs low (*n* = 581) levels *RNF40* (probe 206845_s_at) (**i**). **j** Immunohistochemistry of lungs from intratracheal (i.t) injection or surgical resection models stained with an anti-RNF20 antibody. Scale bar, 400 μm. **k** Relative staining intensity (H-Score) of RNF20 in i.t and tumor relapse groups (*n* = 5). Statistical analysis between two groups (**f, h, k**) was performed using an unpaired two-tailed Student's t-test. Multiple comparisons were performed using one-way ANOVA with significances shown relative to NL or between the indicated pairs of AD grades (**e**). Data are shown as mean ± SEM. 'n' indicates biological replicates.

repair[27,28]. In addition, RNF20-mediated H2B monoubiquitination (H2Bub1) facilitates transcriptional elongation in concert with the FACT complex, by destabilizing nucleosomes[29]. On the other hand, RNF20 hinders the recruitment of TFIIS, which is required for the release of RNA polymerase II (Pol II) into active elongation at tumor-promoting genes, thereby suppressing a pro-oncogenic transcriptional program[30]. Loss of RNF20 and decreased H2Bub1 have been documented in different types of cancer and are associated with an aggressive phenotype[31]. However, the molecular mechanisms that link RNF20 to tumor development and progression remain insufficiently understood.

Here, we show that *Rnf20* haploinsufficiency leads to the spontaneous formation of lung tumors, showing similar morphology and histopathology to human adenocarcinoma and small cell lung carcinoma (SCLC). Mechanistically, *Rnf20* loss results in pronounced defects in DNA repair, increased cell growth, epithelial-mesenchymal transition (EMT), and metabolic rewiring induced by HIF1α activation. *RNF20* levels correlate with the expression of HIF1α and its target genes in lung cancer patients, and *Hif1a* silencing or inhibition of glucose uptake rescue the increased growth and invasion capacity induced by RNF20 deficiency, suggesting that these approaches may be valuable therapeutic strategies in tumors with decreased RNF20 activity.

## Results

### Rnf20 haploinsufficiency leads to the spontaneous formation of lung tumors

In previous work, genetic loss of function of *Rnf20* was shown to predispose mice to acute and chronic colon inflammation, as well as inflammation-associated colorectal cancer[32]. This study used a gene trap insertion within the *Rnf20* gene, predicted to result in a protein that lacks the C-terminal RING domain, which is essential for its E3 ligase activity. We generated a mouse line carrying a LacZ insertion in the second intron resulting in a full null allele[33]. In agreement with the previous investigation, no viable *Rnf20*-/- offspring was detected (Supplementary Fig. 1a–c). *Rnf20* +/- mice, which showed lower *Rnf20* mRNA and protein levels (Supplementary Fig. 1d, e) appeared normal (Supplementary Fig. 1f). Intriguingly, at 1 year of age we detected a high incidence of spontaneous lung tumor formation. 24 out of 32 *Rnf20* +/- mice developed spontaneous tumors, while only 1 out of 20 control wild-type littermates had developed spontaneous tumors by this age (Fig. 1a–c, Supplementary Fig. 1g). 56% of the lesions in *Rnf20* +/- heterozygous mice showed similar morphological and immunohistochemical characteristics to small cell lung cancer (SCLC), i.e., small cells with scant cytoplasm and densely packed chromatin, which were positive for the calcitonin gene-related peptide (CGRP) and synaptophysin (Syp), markers for small cell lung cancer of neuroendocrine origin[34] and negative for CD45, a marker of nucleated hematopoietic cells (Fig. 1a–c, Supplementary Fig. 1h). Apart from tumors with an SCLC phenotype, we also detected tumors in the distal parts of the lungs exhibiting features typical for adenocarcinoma (AD). Immunoreactivity for the lung adenocarcinoma markers thyroid transcription factor-1 (TTF1) and Napsin A, together with negative staining for the squamous cell carcinoma marker p63 confirmed that these lesions are adenocarcinomas (Fig. 1c, middle and bottom panels).

Together, the absence of a single functional *Rnf20* allele is sufficient to trigger spontaneous lung tumor formation. Consistent with this, we found significantly decreased RNF20 protein levels in patients with SCLC and a progressive loss in high-grade adenocarcinoma AD (Fig. 1d, e). Similarly, analysis of TCGA data indicated that in human lung tumors also the *RNF20* mRNA levels are decreased in AD patients (Fig. 1f). Moreover, lower RNF20 levels strongly correlated with poor survival in the KMplotter lung adenocarcinoma dataset[35] (Fig. 1g). By contrast, we did not observe changes in *RNF20* mRNA levels or association with patient survival in squamous cell carcinoma patients (Supplementary Fig. 1i, j).

RNF20 or RNF40 form a heterodimer that acts as the major E3 ligase responsible for histone H2Bub1 in mammalian cells[26]. Interestingly, in contrast to *RNF20*, *RNF40* mRNA levels were increased in AD patients (Fig. 1h) and higher *RNF40* levels significantly correlated with poor survival in the KMplotter lung adenocarcinoma dataset[35] (Fig. 1i), suggesting distinct function of these two proteins in lung cancer.

To further investigate the relationship between RNF20 levels in primary tumors and metastases, we analyzed lung sections from two experimental models[36]. In the first model, mice received an intratracheal injection of LLC1 cells. In the second, a spontaneous metastasis model with tumor resection (surgical resection model), LLC1 cells were injected subcutaneously to establish a primary tumor, which was surgically removed after 10 days to allow for metastasis relapse. Importantly, RNF20 levels were significantly higher in primary tumors, such as those from the intratracheal injection model, compared to tumors in the surgical resection model (Fig. 1j, k).

### Rnf20 haploinsufficiency leads to marked defects in DNA repair and downregulation of p53 and Rb

To determine the molecular mechanisms leading to spontaneous AD and SCLC formation caused by *Rnf20* haploinsufficiency, we first analyzed RNF20 expression in mouse and human lung epithelial and lung cancer cell lines. Lower levels of RNF20 were observed in A549, A427 and H322 adenocarcinoma as well as human H82 and H69 small-cell lung cancer cells compared to normal human bronchial epithelium (BEAS-2B) cells (Fig. 2a) and in mouse Lewis lung carcinoma (LLC1) cells compared to mouse lung epithelial (MLE12) cells (Fig. 2b).

To study the early oncogenic events resulting from *Rnf20* heterozygous loss, we ablated *Rnf20* in MLE12 lung epithelial cells using CRISPR-Cas9 gene editing (Fig. 2c, d and Supplementary Fig. 2a). We were not able to establish homozygous *Rnf20* deletion in any of the cell

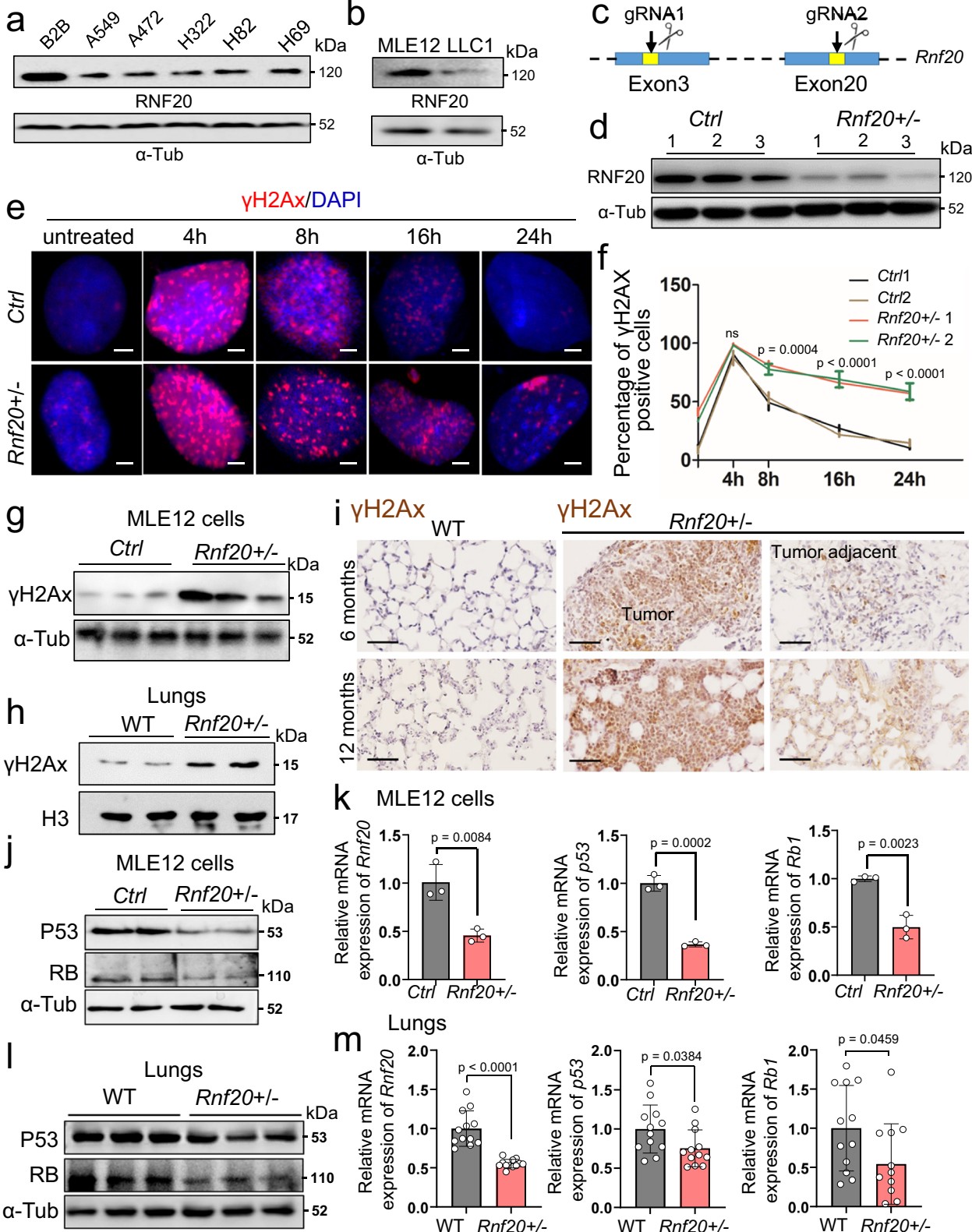

lines mentioned above, suggesting that Rnf20 is essential for cell viability. Importantly, *Rnf20 +/-* MLE12 cells showed an increased number of loci positive for γH2AX, a marker of double-strand breaks, without any DNA damage-inducing agents (Fig. 2e, f). Treatment with $H_2O_2$ induced a major increase in γH2AX loci in both control and *Rnf20 +/-* MLE12, and staining of γH2AX at different time points after treatment revealed pronounced defects of DNA repair in *Rnf20 +/-* cells

(Fig. 2e, f), in line with knockdown studies in HeLa cells[27,28]. Consistent with the immunostainings, we observed higher levels of γH2AX in protein extracts from *Rnf20 +/-* MLE12 cells and mouse lungs (Fig. 2g, h). Immunohistochemical analysis revealed high levels of γH2AX in tumors and adjacent normal lung tissue in *Rnf20 +/-* mice, whereas we hardly detected any γH2AX-positive cells in wild-type lungs, supporting a role RNF20 in safeguarding genomic stability

**Fig. 2 | *Rnf20* haploinsufficiency results in pronounced defects in DNA repair and decrease in p53 and RB levels. a** Western blot analysis of total cell lysates of human bronchial epithelium B2B (BEAS-2B), A549, A427 and H322 adenocarcinoma as well as human H82 and H69 small-cell lung cancer cells. α-Tubulin served as a loading control. **b** Western blot analysis of total cell lysates of mouse lung epithelial (MLE12) cells and mouse Lewis lung carcinoma (LLC1) with RNF20 antibody. **c** Schematic representation of the strategy used to generate *Rnf20* +/- MLE12 cells. The scissors icon is created in BioRender[67]. **d** Western blot analysis of total cell lysates of control (*Rnf20* +/+) and *Rnf20* +/- MLE12 cells. **e** Costaining for γH2AX (red) and DAPI (blue) of wild-type and *Rnf20* +/- MLE12 cells at the indicated time points after treatment with 0.5 mM H2O2. Scale bar, 4 μm. **f** Kinetics of γH2AX foci

from (**e**, *n* = 3). **g, h** Western blot analysis for γH2AX of total cell lysates of control and *Rnf20* +/- MLE12 cells (**g**) and isolated lungs from control wild-type littermates and *Rnf20* +/- mice (**h**). **i** Immunohistochemical staining of wild-type and *Rnf20* +/- mice at 6 months and 1 year of age with γH2AX. Scale bar, 200 μm. **j, l** Western blot analyses for P53, RB1 and α-Tubulin of total cell lysates of control and *Rnf20* +/- MLE12 cells (**j**), or wild-type and *Rnf20* +/- lungs (**l**). **k, m** Quantitative PCR (qPCR) analysis of *Rnf20*, *p53 (TrpS3)* and *Rb1* expression in control and *Rnf20* +/- MLE12 cells (**k**, *n* = 3), or wild-type (*n* = 12) and *Rnf20* +/- (*n* = 12) lungs (**m**). Statistical analysis between two groups (**f, k, m**) was performed using an unpaired two-tailed Student's t-test. Data are shown as mean ± SEM. 'n' indicates biological replicates.

(Fig. 2i). The p53 and retinoblastoma (RB) tumor suppressor pathways control the DNA damage response[13,37] and alterations in these genes are among the most common genetic changes in lung cancer. Acute silencing of *RNF20* by RNAi in HeLa cells results in a major decrease of p53 expression[38], while the effect on RB is not known. We observed a significant decrease in mRNA and protein levels of *Trp53* (*p53*) and *Rb1* in *Rnf20* haploinsufficient MLE12 cells and lungs (Fig. 2j–m), suggesting that increased DNA damage and insufficient activation of key tumor suppressor pathways may contribute to the increased incidence of lung tumorigenesis upon *Rnf20* depletion.

## Rnf20 haploinsufficincy leads to epithelial-mesenchymal transition (EMT) and increased HIF1α signaling

Another hallmark of cancer progression is invasion and metastasis[14], which is often preceded by activation of an epithelial-mesenchymal transition (EMT) program[39]. Importantly, while control MLE12 cells maintained a cobblestone-like (epithelial) appearance in culture, *Rnf20* +/- MLE12 cells showed irregular spindle-shaped morphology (Fig. 3a). We detected increased lamellipodia and filopodia via F-actin immunostaining (Fig. 3b), as well as significant downregulation of the epithelial marker E-cadherin, and upregulation of the mesenchymal markers fibronectin (Fn1), vimentin (Vim) and N-cadherin in *Rnf20* +/- cells (Fig. 3c, d and Supplementary Fig. 2b, c). In line with the morphological alterations, *Rnf20* +/- MLE12 cells possessed a significantly higher migratory capacity in scratch wound healing and Boyden chamber-based migration assays (Fig. 3e, and Supplementary Fig. 2d, e).

Next, we studied the effect of *Rnf20* haploinsufficiency on cell proliferation in adherent 2D monolayer cultures and on the capacity to form colonies in soft agar as a measure of anchorage-independent growth, hallmark of malignant transformation[14]. *Rnf20* deficiency resulted in markedly increased colony formation in monolayer and soft agar (Fig. 3f and Supplementary Fig. 2f). To determine whether the observed effects are a functional consequence of RNF20 loss, we next overexpressed RNF20. This overexpression reduced the elevated colony formation and migration abilities of *Rnf20* +/- cells to levels comparable to those of control cells (Supplementary Fig. 2g, h).

To further dissect the underlying mechanisms, we performed RNA-sequencing of control and *Rnf20* +/- MLE12 cells. We identified 1142 genes significantly upregulated and 844 downregulated in *Rnf20* haploinsufficient cells (Supplementary Data 1, Log2(FC) ≤−0.58, ≥0.58; p-value < 0.05). Gene Ontology (GO) analysis of genes upregulated in *Rnf20* +/- MLE12 cells revealed over-representation of GO terms linked to regulation of tube morphogenesis, extracellular matrix (ECM) organization, HIF1 signaling pathway, insulin resistance as well as mesenchymal differentiation and positive regulation of cell migration (Fig. 3g, h). Top downregulated genes were linked to the TNF signaling pathway and drug metabolism. Quantitative PCR analysis and immunostainings of control and *Rnf20* +/- lungs confirmed the decrease of the epithelial marker E-cadherin and increase of the mesenchymal marker *Fn1* as well as upregulation of *Snai1*, a master regulator of EMT (Fig. 3i, j).

## HIF1α activation in Rnf20 +/- cells results in metabolic rewiring and increased DNA damage, cell growth, migration and invasion

KEGG pathway enrichment analysis confirmed the HIF1α-pathway as a top pathway upregulated in *Rnf20* haploinsufficient cells (Fig. 4a). Therefore, we next assessed HIF1α levels in control and *Rnf20* +/- MLE12. We observed a dramatically augmented accumulation of HIF1α in *Rnf20* haploinsufficient cells in both normoxic and hypoxic conditions (1% oxygen, Fig. 4b), while *Hif1a* mRNA levels did not increase (Supplementary Fig. 3a). Consistent with the increased protein levels of HIF1α we observed significant increase in HIF1α reporter gene activity and upregulation of a large set of enzymes involved in glycolysis, which are also HIF1α target genes (Fig. 4c, d, Fig. 3g, h). Liquid chromatography–mass spectrometry (LC–MS) analysis revealed a concomitant increased production of a series of glycolytic intermediate metabolites (Fig. 4d, Supplementary Data 2). Extracellular acidification rate (ECAR) measurements confirmed the significantly increased glycolytic flux and increased maximum glycolytic capacity of *Rnf20* +/- MLE12 cells (Fig. 4e).

Since the glucose transporter GLUT1 (*Slc2a1*) that facilitates the uptake of glucose from the extracellular medium was upregulated in *Rnf20* +/- MLE12 cells, we measured glucose uptake directly using the glucose analog 2-deoxyglucose (2-DG), which cannot be further catabolized. Indeed, 2-DG uptake was significantly increased in *Rnf20* +/- MLE12 cells compared to controls (Fig. 4f). Another markedly upregulated enzyme was lactate dehydrogenase A (*Ldha*), which catalyzes the last step of anaerobic glycolysis, producing lactate (Fig. 4d and Supplementary Fig. 3b). We detected a significant increase of cellular lactate levels (Fig. 4d), as well as lactate in the cell culture medium of *Rnf20* +/- MLE12 cells compared to controls (Fig. 4g). In addition, we detected an increase in TCA metabolites, which was less pronounced compared to the increase of lactate (Fig. 4d). This might be due to the increase of pyruvate dehydrogenase kinases (PDKs, Fig. 3h and Supplementary Fig. 3b), mitochondrial enzymes inhibiting the pyruvate dehydrogenase complex that converts cytosolic pyruvate to mitochondrial acetyl-CoA, which is the first substrate for the TCA cycle. Importantly, we did not observe increased mitochondrial reactive oxygen species (ROS) production, suggesting that HIF1α stabilization is not dependent on increased mitochondrial ROS or total ROS in *Rnf20* +/- cells (Supplementary Fig. 3c–e).

We next analyzed the levels of key enzymes involved in glucose metabolism in isolated lungs of *Rnf20* +/- mice and wild-type littermates. We detected a significant increase in *Slc2a1*, *Ldha*, *Pdk1*, and *Eno1* mRNA levels (Fig. 4h) and increased GLUT1, LDHA, and HIF1α in immunohistochemical stainings (Fig. 4i, j). Consistent with the increased glucose uptake and increased levels of secreted lactate in the cell culture system, we observed elevated glucose-6-phosphate and fructose-6-phosphate in lung homogenates and lactate levels in blood serum of *Rnf20* +/- compared to control mice (Fig. 4k), underscoring the in vivo relevance of our findings.

Taken together, the above findings demonstrated strong nuclear accumulation of HIF1α coupled with a robust increase in glycolysis as evidenced by an increase in glucose uptake, glycolytic flux, and lactate production, suggesting that HIF1α activation might be

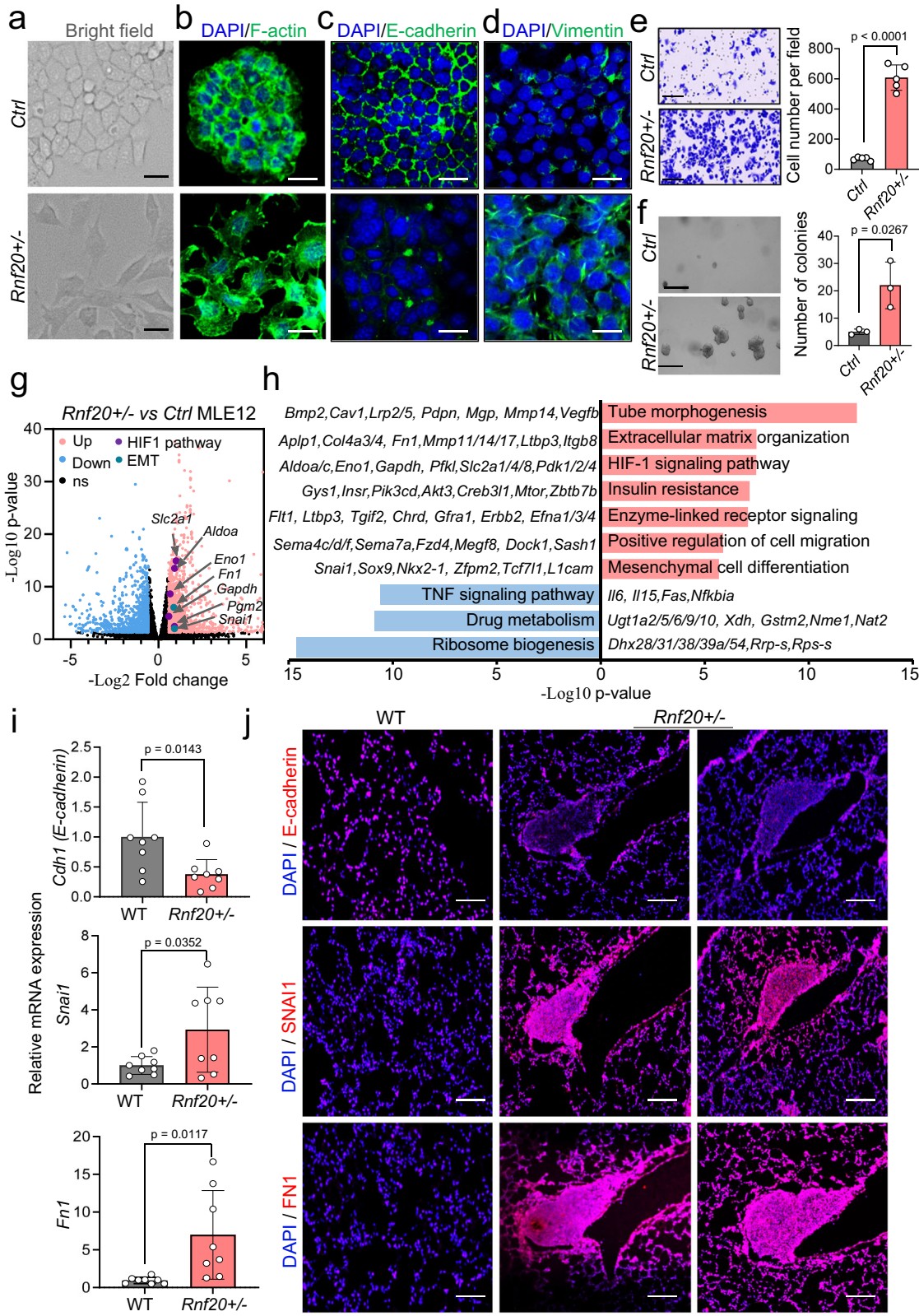

responsible for the increased lung tumorigenesis in *Rnf20 + /-* mice. Thus, we next studied whether shRNA-mediated silencing of *Hif1a* could rescue the metabolic changes, increased growth, and migration of *Rnf20* haploinsufficient cells. Indeed, silencing of *Hif1a* decreased the expression of HIF1α-dependent genes, the number of colonies in both 2D culture conditions and soft agar, as well as the migration capacity of *Rnf20 + /-* MLE12 cells, to control levels (Fig. 5a–c,

Supplementary Fig. 4a–d). Furthermore, glucose uptake, lactate secretion, and glycolytic flux were reduced to control levels in *Hif1a*-depleted *Rnf20 + /-* cells (Fig. 5d–f), supporting a critical role of HIF1α in mediating the phenotype elicited by *Rnf20* haploinsufficiency. Moreover, γH2AX was increased under hypoxia and RNF20 over-expression or *Hif1a* silencing was sufficient to reduce HIF1α and γH2AX levels in *Rnf20 + /-* MLE12 cells to control levels (Fig. 5g and

**Fig. 3 | Rnf20 haploinsufficiency results in epithelial-mesenchymal transition (EMT) and altered HIF1α signaling. a** Bright-field images of control and *Rnf20 +/-* MLE12 cells. Scale bars, 50 μm. **b, c, d** Phalloidin stainings for F-actin (**b**) and immunostaining using E-cadherin (**c**), and vimentin antibodies (**d**) on control and *Rnf20 +/-* MLE12 cells. Cell nuclei were labeled with DAPI. Scale bars, 25 μm. **e** Boyden chamber migration assay with control and *Rnf20 +/-* MLE12 cells (left) and quantification of the number of migrated cells per field (right). *n* = 5. Scale bars, 200 μm. **f** Soft agar assay to determine the anchorage-independent cell growth of control and *Rnf20 +/-* MLE12 cells. Bright-field images (left) show the increased colony size of *Rnf20* haploinsufficient cells, and the graph (right) shows quantification of the number of colonies (*n* = 3). Scale bars, 200 μm. **g** Volcano plot showing the distribution of differentially expressed genes for *Rnf20 +/-* versus control MLE12 cells. *n* = 3; Log2(FC) ≤−0.58, ≥0.58; p-value < 0.05. Differential expression analysis was performed using DESeq2 (Version 1.40.0). **h** Top 10 GO terms of genes upregulated and downregulated upon RNF20 haploinsufficiency. Representative genes are shown next to the GO terms. GO term enrichment analysis was performed using Metascape (Version 3.5). **i** RT-qPCR validation of EMT genes in RNA isolated from control (*n* = 8) and *Rnf20 +/-* (*n* = 8) mouse lungs. mRNA expression is presented relative to control wild-type littermates. **j** Immunostaining for E-cadherin, SNAI1 and FN1 in lung tissues sections of control and *Rnf20 +/-* mice. Scale bars, 200 μm. Statistical analysis between two groups in (**e, f, i**) was performed using an unpaired two-tailed Student's t-test. Data are shown as mean ± SEM. 'n' indicates biological replicates.

---

Supplementary Fig. 4e). Moreover, overexpression of constitutively active (non-degradable) human HIF1α, similar to RNF20 loss, led to enhanced clonogenic growth, and elevated expression of its targets, *Slc2a1* and *Ldha* (Supplementary Fig. 4f–h). Collectively, these data suggest that HIF1α activation, together with insufficient DNA repair, might be responsible for the phenotypic changes upon *Rnf20* haploinsufficiency.

Since we detected a major upregulation of the glucose transporter GLUT1 (*Slc2a1*) in *Rnf20* haploinsufficient cells, we next tested whether inhibition of glucose uptake would reduce the increased proliferation and migration capacity of *Rnf20 +/-* cells. Treatment with the GLUT1 inhibitor WZB117 decreased the glucose uptake, glycolytic flux and lactate secretion in both control and *Rnf20 +/-* MLE12 cells (Fig. 5h–j, Supplementary Fig. 4i). Importantly, the number of colonies in both 2D culture conditions and soft agar assay, as well as the migration capacity, of *Rnf20 +/-* MLE12 cells was greatly decreased after WZB117 treatment (Fig. 5k, l and Supplementary Fig. 4j).

Recently, the RNF20-RNF40 complex was shown to be required for HIF-1 transcriptional activity in breast cancer cells using double knockdown experiments[40]. Given our finding that RNF20 suppresses HIF1α activity, we next silenced *Rnf40* in MLE12 cells. In contrast to RNF20 LOF, *Rnf40* silencing did not affect HIF1α levels (Fig. 5m), cell migration or clonogenic growth (Fig. 5n, o and Supplementary Fig. 4k, l), but significantly reduced the expression of some HIF1α targets, such as *Fn1*, *Snai2*, *Vegfa* and *Eno1* (Fig. 5p). These findings indicate that RNF20 operates independently of RNF40 in suppressing cell growth, migration, and the regulation of HIF1α-responsive genes, at least in lung epithelial cells and lung cancer.

### RNF20 Haploinsufficiency drives tumor growth and metastasis via HIF1α activation and metabolic rewiring

Next, we investigated the impact of *Rnf20* loss on lung cancer development and progression. Consistent with findings in MLE12 cells, we observed that the ablation of a single *RNF20* allele in the human A549 adenocarcinoma cell line led to increased levels of HIF1α and γH2AX (Fig. 6a, b). Further, RNF20 loss was sufficient to enhance clonogenic growth, migration, glucose uptake, glycolysis, and lactate secretion, effects that could all be suppressed by HIF1A silencing (Fig. 6c–h). Moreover, inhibition of glucose uptake significantly reduced the enhanced clonogenic growth and migration abilities of *RNF20 +/-* A549 cells (Fig. 6i, j). Consistent findings were observed with siRNA-mediated silencing of *RNF20* in the human small-cell lung cancer cell line H82 (Fig. 6k–m), as well as with the ablation of a single *Rnf20* allele in murine Lewis lung carcinoma (LLC1) cells (Supplementary Fig. 5a–k).

To further investigate the role of RNF20 loss in tumor growth and metastatic dissemination in vivo, we injected control and *RNF20 +/-* A549 cells subcutaneously and intravenously into nude mice. Tumor volume in mice transplanted with *RNF20 +/-* A549 cells was already larger 9 days post-injection and continued to increase thereafter (Fig. 7a, b). Similarly, intravenous injection of *RNF20 +/-* A549 cells resulted in a significantly higher number of tumor nodules and an expanded metastatic area in the lungs compared to mice injected with control A549 cells (Fig. 7c, d).

Consistent with these findings, C57BL/6 mice injected intravenously with *Rnf20 +/-* LLC1 cells also exhibited a significantly increased number of pulmonary tumor nodules and a larger metastatic area compared to controls (Fig. 7e, f).

Importantly, in line with our cell culture-based assays, *HIF1A* silencing or inhibition of glucose uptake with the GLUT1 inhibitor WZB117 was sufficient to suppress the metabolic changes (Fig. 7g) and reduce both the number of tumor nodules and the metastatic area in mice injected with *RNF20 +/-* A549 cells (Fig. 7h–j). Notably, tumors from mice injected with *RNF20 +/-* A549 cells exhibited significantly elevated γH2AX levels compared to those developed in mice injected with control A549 cells, an effect that was reversed by *HIF1A* silencing or glucose uptake inhibition (Fig. 7h, k). Taken together, these data suggest a critical role of HIF1α-mediated metabolic rewiring in the increased tumor growth and metastasis upon RNF20 LOF in vivo.

### RNF20 controls Rbx1-mediated HIF1α degradation

RNF20 functions as an E3 ubiquitin ligase for H2Bub1. To investigate the mechanisms underlying the role of RNF20 in HIF1α activation and metabolic rewiring during tumorigenesis, we next examined how H2Bub1 enrichment on chromatin correlates with transcriptional changes upon *Rnf20* LOF. Genome-wide H2Bub1 levels were decreased in *Rnf20 +/-* MLE12 cells (Fig. 8a, b), to a similar extent as observed at genes upregulated in *Rnf20* haploinsufficient cells, while the relative decrease of H2Bub1 levels was more prominent at downregulated genes, which showed higher H2Bub1 enrichment in control cells (Fig. 8b, c). Among genes that showed decreased H2Bub1 levels coupled to decreased expression were the known RNF20 target *p53*, as well as genes involved in the Notch signaling pathway and HIF1α degradation, such as *Hes1* and *Rbx1* (Fig. 8b). RBX1 is a component of the VHL tumor suppressor complex, which ubiquitinates HIFα subunits (HIF1α, HIF2α, and HIF3α) and targets them for degradation under normoxic conditions[41,42]. Interestingly, *Rbx1* was specifically downregulated upon RNF20 LOF, but not following RNF40 depletion, which did not alter Hif1α levels (Supplementary Fig. 6a). To investigate whether decreased RBX1 levels are responsible for the elevated HIF1α levels observed upon *Rnf20* loss, we overexpressed RBX1 in *Rnf20 +/-* cells. RBX1 overexpression led to a significant reduction in HIF1α levels (Fig. 8d), reduced expression of HIF1α target genes involved in glycolysis (Fig. 8e), and decreased clonogenic growth and migration capacity of *Rnf20 +/-* cells (Fig. 8f, g). Taken together, these data suggest that the reduction in RBX1 levels caused by *Rnf20* LOF is responsible for the increased HIF1α levels and activity under normoxic conditions.

### HIF1α activation upon Rnf20 haploinsufficiency induces RNA polymerase II promoter-proximal pause release at metabolic genes

HIF1α was shown to modulate gene expression programs controlled by RNA polymerase II (Pol II) pausing[43] and similar findings have been

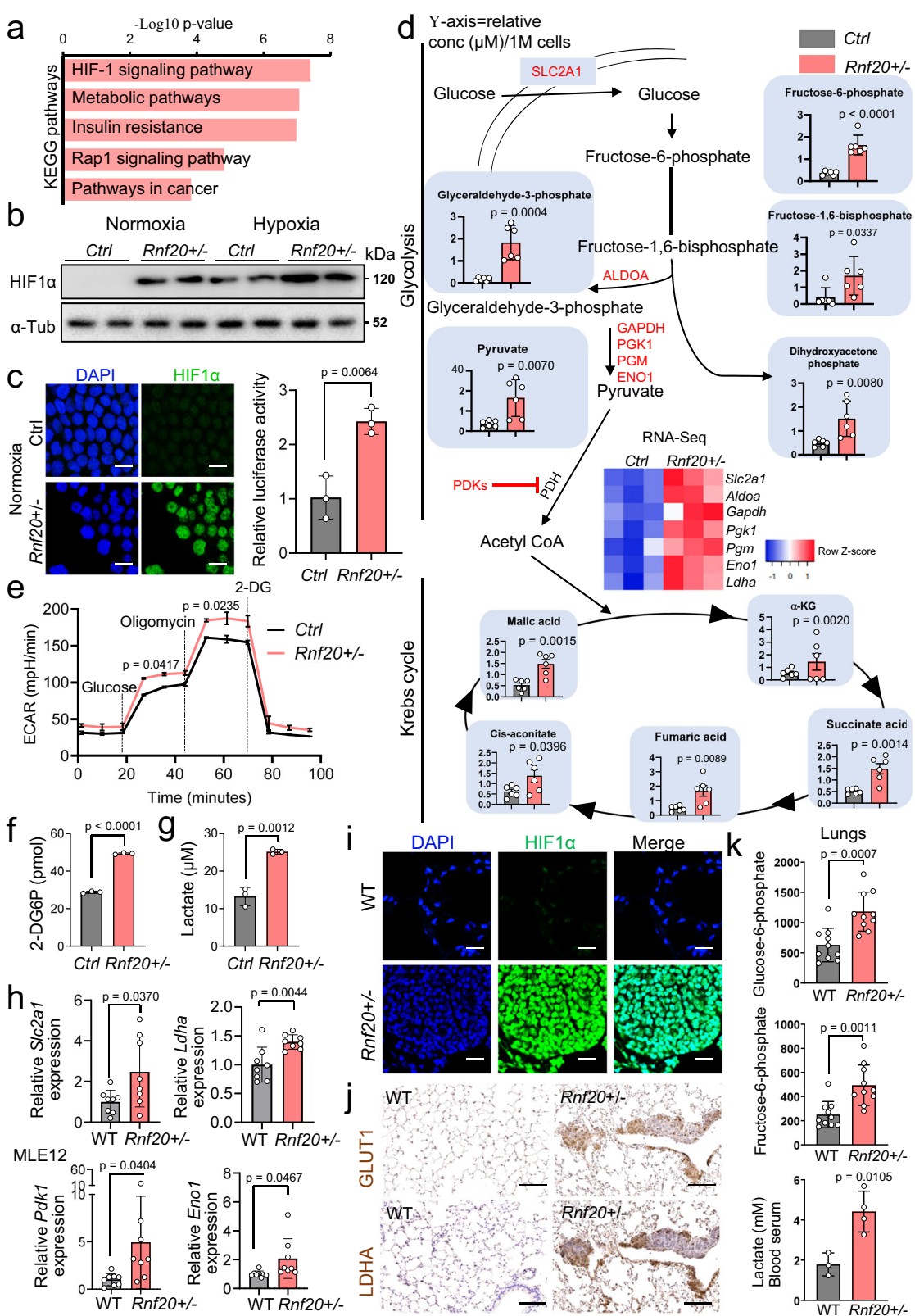

reported for RNF20[30]. Therefore, we next performed ChIP-Seq for total Pol II in control and *Rnf20 + /-* MLE12 cells and determined the pausing index, as calculated by the Pol II binding at the promoter-proximal region versus the gene body (Fig. 9a and Supplementary Data 3). We found a significant decrease in the pausing index at genes upregulated in *Rnf20 + /-* versus control MLE12 cells, while we did not detect major changes at genes that were not changed or downregulated (Fig. 9b, c

and Supplementary Fig. 6b). Genes that showed a decreased pausing index and upregulation in *Rnf20* haploinsufficient cells, were linked to glycolysis, cell cycle control and metabolism as well as autophagy and cytoskeleton organization (Fig. 9d, e and Supplementary Data 4), many of which are regulated by HIF1α or a linked to EMT. In line with this, the pausing index at EMT and HIF1α-pathway genes was also decreased in *Rnf20 + /-* compared to control MLE12 cells (Fig. 9f). Pol II ChIP-qPCR

**Fig. 4 | Metabolic rewiring upon *Rnf20* haploinsufficiency. a** KEGG pathway analysis of genes upregulated in *Rnf20 +/-* MLE12 cells compared to controls. Significance is presented as -Log10 (p-values) from KEGG pathway analysis using DAVID Bioinformatics Resources. **b** Western blot analysis for HIF1α in control and *Rnf20 +/-* MLE12 cells under normoxic or hypoxic conditions. **c** Immunostaining for HIF1α (left panels) and HIF1α luciferase reporter activity (right panel) in control and *Rnf20 +/-* MLE12 cells (*n* = 3) under normoxic conditions. Scale bars, 20 μm. **d** Targeted LC-MS/MS-based metabolomics of control and *Rnf20 +/-* MLE12 cells (*n* = 6). The *Y*-axis represents the relative concentration (μM) per 1 million cells. Metabolic enzymes that show changed expression in *Rnf20 +/-* MLE12 cells are shown in red. The heatmap in the middle shows the gene expression changes in control and *Rnf20 +/-* MLE12 cells. **e** Extracellular acidification rate (ECAR) of control and *Rnf20 +/-* MLE12 cells (*n* = 3) recorded following the addition of glucose, oligomycin, and 2-DG. **f, g** 2-DG uptake (**f**) and concentration of lactate in the supernatant (**g**) of control and *Rnf20 +/-* MLE12 cells (*n* = 3). **h** qPCR analysis of key glycolytic enzymes and hypoxia-regulated genes in control (*n* = 8) and *Rnf20 +/-* (*n* = 8) lungs. **i, j** Immunostaining for HIF1α (**i**), and immunohistochemistry for GLUT1 and LDHA in lung tissues sections (**j**) of wild-type and *Rnf20 +/-* mice. Scale bars in (**i**) 100 μm; in (**j**) 200 μm. **k** Concentration of glucose-6-phosphate and fructose-6-phosphate in lung homogenates (top and middle panels, *n* = 10 each), and lactate in the serum (bottom panel) of wild-type (*n* = 3) and *Rnf20 +/-* (*n* = 4) mice. Statistical analysis between two groups in (**c, d, e, f, g, h, k**) was performed using an unpaired two-tailed Student's t-test. Data are shown as mean ± SEM. 'n' indicates biological replicates.

on glycolytic targets in control and *Rnf20 +/-* MLE12 cells as well as in WT and *Rnf20 +/-* lung samples, confirmed the decreased Pol II pausing upon *Rnf20* loss, both in vitro and in vivo (Supplementary Fig. 6c and Fig. 9g).

Since HIF1α has been shown to stimulate Pol II pause release, we next intersected HIF1α-bound genes in A549 cells[44] with genes exhibiting decreased PI upon RNF20 loss. GO analysis revealed that the overlapping genes are associated with the mRNA metabolism, HIF-1 signaling pathway, glycolytic processes and mechanisms associated with pluripotency (Fig. 9h). Next, we performed Pol II ChIP-Seq in control, *Rnf20 +/-,* and *Rnf20 +/-* HIF1α knockdown MLE12 cells. Consistent with a direct role of HIF1α in regulating Pol II pause release, the reduced Pol II pausing index in *Rnf20 +/-* MLE12 cells at HIF1α-bound genes was increased to control levels upon HIF1α depletion (Fig. 9i, j). To validate these findings, we performed ChIP-qPCR for initiating/paused and elongating Pol II enriched in phosphorylated serine 5 (pSer5) and serine 2 (pSer2), respectively, on glycolytic targets in control, *Rnf20 +/−*, and *Rnf20 +/−* HIF1α knockdown MLE12 cells. We observed increased enrichment of elongating Pol II-pSer2 across the gene bodies of *Slc2a1 (Glut1)*, *Eno1*, and *Ldha* upon RNF20 loss, which was attenuated upon HIF1α depletion (Fig. 9k). Concurrently, initiating/paused Pol II-pSer5 was reduced at the transcription start sites of these genes in *Rnf20 +/−* MLE12 cells, but was restored to control levels following HIF1α depletion (Fig. 9k).

RNF20 functions as an E3 ubiquitin ligase for H2Bub1, a modification that promotes SET1-dependent di- and trimethylation of H3K4[45]. SET1B is also required for the activation of HIF-inducible genes[46]. To investigate the role of RNF20 and HIF1α on transcriptional regulation in lung epithelial cells, we analyzed the relationship between H2Bub1 changes and PI in *Rnf20 +/-* versus control MLE12 cells and performed ChIP-Seq for H3K4me3 in *Rnf20 +/-* and *Rnf20 +/-* HIF1α knockdown MLE12 cells. The results showed no significant correlation between PI and H2Bub1 changes, indicating that Pol II pausing is independent of H2Bub1 levels (Supplementary Fig. 6d). Furthermore, we did not observe changes in H3K4me3 levels at HIF1α target genes or genes upregulated in *Rnf20 +/-* cells upon HIF1α silencing, but we identified a decrease in H3K4me3 at genes downregulated by RNF20 LOF (Supplementary Fig. 6e). These findings suggest that the HIF1α-dependent reduction of H3K4me3 may contribute to the transcriptional downregulation of certain RNF20 target genes. However, this mechanism does not appear to play a role in the regulation of RNF20-dependent transcriptional pausing at HIF1α target genes.

**HIF1α and HIF1α-target expression correlate with RNF20 levels in lung cancer patients**

Our experiments demonstrated that HIF1α-mediated reprogramming plays a major role in mediating the phenotype elicited upon RNF20 loss, therefore, we next correlated *RNF20* levels with the expression of HIF1α and HIF1α-dependent metabolic enzymes in lung cancer patients. We observed significantly increased protein levels of HIF1α, ENO1, and LDHA in patients with SCLC and high-grade AD, which were inversely correlated with RNF20 protein levels (Fig. 10a, b). Furthermore, *SLC2A1*, *ENO1*, *LDHA* and *PDK1* mRNA levels were high in AD patients (Supplementary Fig. 7a), and we found significant negative correlation between *RNF20* levels and *SLC2A1*, *ENO1*, *LDHA* and *VEGFA* expression (Supplementary Fig. 7b). Moreover, in line with the survival data, showing that decreased *RNF20* mRNA levels are associated with decreased survival only in AD patients but not in SC patients, we observed decreased survival in AD patients expressing high level of *SLC2A1*, *ENO1*, *LDHA* and *VEGFA*, but not in squamous cell carcinoma patients (Supplementary Fig. 7c, d).

## Discussion

Our study uncovers a key role of RNF20 in safeguarding genome integrity as well as inhibiting EMT and metabolic rewiring in lung epithelial and neuroendocrine cells and points to HIF1α inhibition as a potential therapeutic strategy for AD and SCLC patients with decreased *RNF20* function (Fig. 10c).

The most common causes of all lung cancer types are smoking and exposure to other carcinogens, such as radon, asbestos, and air pollution[47]. These substances can directly damage DNA, which contributes significantly to the development and progression of cancer. RNF20 plays an important role in the repair of double-strand breaks (DSBs) via error-prone nonhomologous end-joining (NHEJ) or high-fidelity repair by homologous recombination (HRR), through its main function of promoting H2Bub1[27,28]. H2Bub1 promotes the recruitment of DNA repair factors, such as 53BP1 and BRCA1, facilitating timely repair[28]. We detected profound DNA damage and greatly increased incidence of lung tumors with characteristics of lung AD and SCLC in *Rnf20* haploinsufficient mice, suggesting that non-functional DNA damage repair due to RNF20 loss might be an important factor in the development of lung cancer (Fig. 10c). Furthermore, we observed significantly decreased *RNF20* expression in patients with SCLC and lung AD. Interestingly, arsenite, a carcinogenic substance occurring in the air, water, and soil, was shown to bind to the RING-containing catalytic domain of the RNF20-RNF40 complex and to inhibit its DSB repair function[48]. A marked increase in bladder and lung cancer mortality rates has been reported due to arsenic in drinking water[49], suggesting that pollution could also cause a LOF of RNF20, with or without changes in gene expression levels, thus contributing to carcinogenesis. Surprisingly, RNF40 was increased in patients with lung AD, and higher levels correlated with poor prognosis. Notably, RNF40 depletion had minimal impact on the cell growth and migration of lung epithelial cells, suggesting that RNF20 and RNF40 have distinct, independent roles in lung cancer development and progression.

There are two primary histopathological groups of lung tumors: SCLC and non-small-cell lung cancer (NSCLC). The latter accounts for approximately 80% of all cases, with adenocarcinoma being the most common type[47]. These different lung cancer types have distinct origins, clinical and pathological features[50]. SCLC is an aggressive type of lung cancer with neuroendocrine origin that typically grows and spreads rapidly (Fig. 10c). Alveolar type II (AT2) cells, which play a

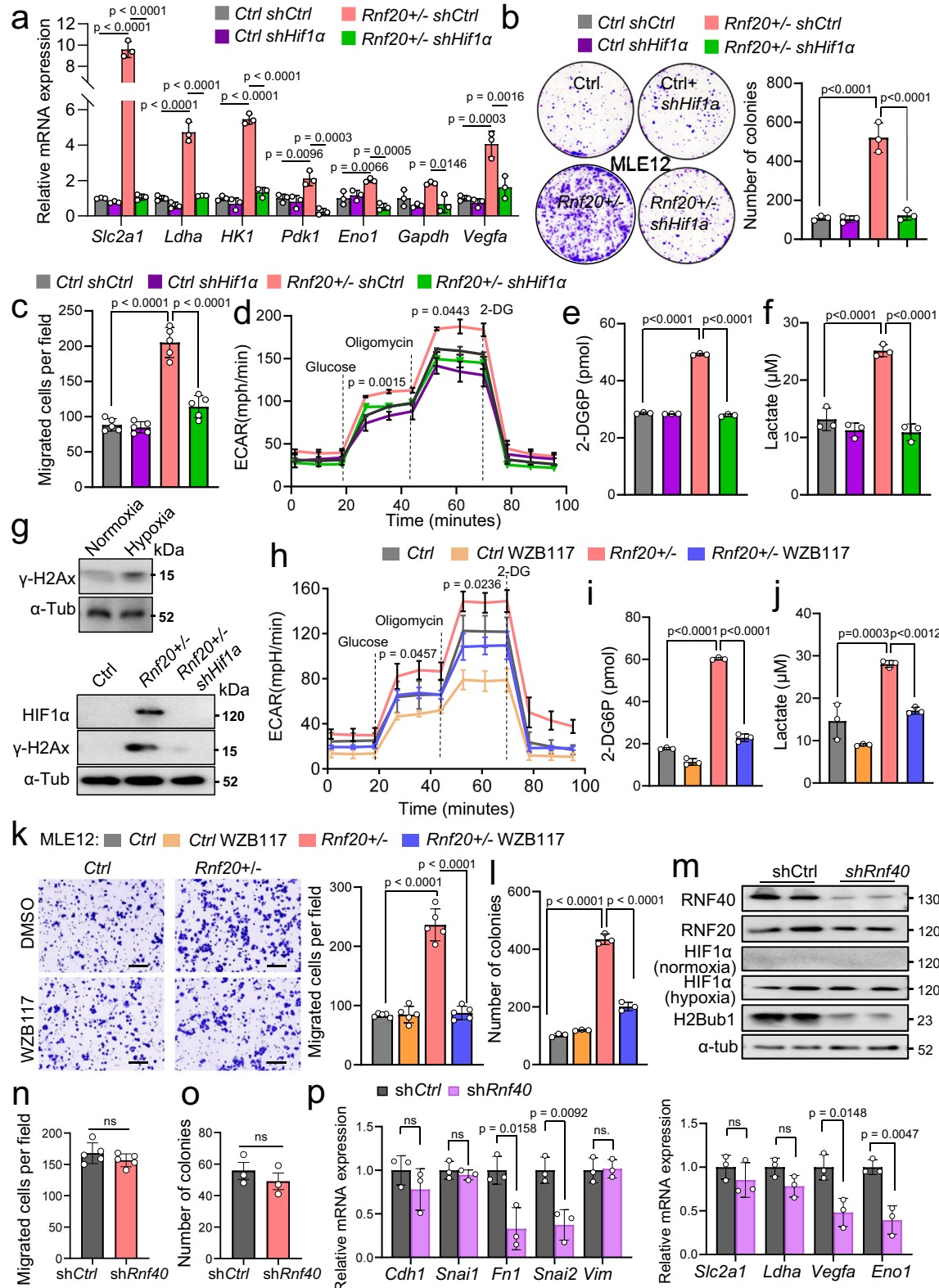

central role in maintaining the integrity and function of the alveoli (Fig. 10c), are the primary cells of origin of lung AD. While adenocarcinoma is the most common subtype, the marked increase in SCLC-like lesions could be attributed to the concomitant decrease in p53 and RB, along with elevated HIF1α activity and decreased Notch signaling. Somatic inactivation of *Rb1* and *p53* using intratracheal injection or intubation of mice with adenovirus-Cre leads to the

development of SCLC-like tumors[34], whereas mice carrying germline deletion of one *Rb1* allele and both p53 alleles (*Rb1+/−p53−/−*) develop a variety of primary tumors of neuroendocrine origin as well as focal bronchial neuroendocrine cell hyperplasia[51], showing the critical importance of these factors in preventing transformation and growth of neuroendocrine cells. Intriguingly, *RB1* knockdown in hESC-derived pulmonary neuroendocrine cells (PNEC) produces

**Fig. 5 | HIF1α activation upon *Rnf20* loss results in metabolic rewiring, DNA damage, and increased cell growth and migration. a** qPCR analysis of genes involved in glycolysis and hypoxic response in control and *Rnf20 +/-* MLE12 cells, stably expressing either control shRNA or shRNA against *Hif1a* (*n* = 3). **b** Clonogenic assay (left) and quantification of the number of colonies (right, *n* = 3). **c** Quantification of the number of migrated cells per field in a Boyden chamber migration assay (*n* = 5). **d–f** ECAR (**d**), 2-DG uptake (**e**) and concentration of lactate in the supernatant (**f**) of control and *Rnf20 +/-* MLE12 cells stably expressing control shRNA or shRNA against *Hif1a* (*n* = 3). **g** Western blot analysis for HIF1α and γH2AX in total cell lysates of MLE12 cells under normoxic or hypoxic conditions (1%O₂) (top) and control, *Rnf20 +/-* or *Hif1a*-knockdown *Rnf20 +/-* MLE12 cells in normoxic conditions (bottom). **h–j** ECAR (**h**), 2-DG uptake (**i**), and concentration of lactate in the supernatant (**j**) of control and *Rnf20 +/-* MLE12 cells treated with DMSO or with

WZB117 (*n* = 3). **k** Boyden chamber migration assay with control and *Rnf20 +/-* MLE12 cells treated with DMSO or with WZB117 (left) and quantification (right) (*n* = 5). Scale bars, 150 μm. **l** Clonogenic assay with control and *Rnf20 +/-* MLE12 cells treated with DMSO or with WZB117 (*n* = 3). **m** Western blot analysis of RNF20, RNF40 and H2Bub1 in normoxic conditions, and HIF1α under normoxic or hypoxic conditions, in total cell lysates from control or *Rnf40*-knockdown MLE12 cells. **n, o** Number of migrated cells per field in a Boyden chamber migration assay (**n**, *n* = 5) and of colonies in a clonogenic assay with control and *Rnf40*-knockdown MLE12 cells (**o**, *n* = 3). **p** qPCR analysis of genes involved in EMT (left) or glycolysis (right) in control and *Rnf40*-knockdown MLE12 cells (*n* = 3). Statistical analysis in (**n, o, p**) was performed using a two-tailed Student's t-test. Multiple comparisons in (**a–f, h–k**) were performed using one-way ANOVA with Tukey's multiple comparisons test. Data are mean ± SEM; ns, non-significant. 'n' indicates biological replicates.

CGRP-expressing cells with gene expression characteristics of SCLC[52]. Bi-allelic inactivation of *TP53* and *RB1* have been reported in nearly all SCLC[53], further supporting the notion that LOF of p53 and RB in *Rnf20* haploinsufficient mice could be responsible for the increased incidence of SCLC-like lesions in these mice. p53 is a known primary downstream target of RNF20[38], whereas the effect of RNF20 on RB has so far not been examined. While we found a major decrease in H2Bub1 at genes downregulated in *Rnf20 +/-* cells, including *p53*, we did not observe profound changes in H2Bub1 at the *Rb1* locus (Fig. 8b), suggesting that the decrease of *Rb1* mRNA levels in *Rnf20* haploinsufficient cells and animals is independent of transcriptional regulation via the Rnf20-H2Bub1 axis. Furthermore, we detected a significant RNF20-H2Bub1-dependent decrease in the expression of *Rbx1*, a key component of the VHL-containing ubiquitin ligase complex that initiates the degradation of HIF1α[42]. As a consequence, HIF1α protein, but not mRNA, levels were elevated in RNF20 LOF cells even under normoxic conditions, thereby disrupting oxygen sensing and creating a state of pseudohypoxia. In this context, it is noteworthy that PNECs, which are considered cells of origin for SCLC, play a crucial role as oxygen sensors. Abnormal oxygen sensing, caused by the loss of prolyl hydroxylase domain proteins (PHDs) function, regulating the stability of HIF-1α, leads to PNEC hyperplasia[54]. Moreover, *Hes1* showed significant downregulation, which was accompanied by a decrease in H2Bub1 levels. Similarly, RNF20 is required for Notch-dependent gene expression in endothelial cells[33], and its Drosophila homolog, dBre1, has been shown to regulate the expression of Notch target genes by coupling H2Bub1 to H3K4me3[55]. Notch signaling suppresses SCLC, as genetic and pharmacological inhibition of Notch activity were associated with an increased tumor number[53], suggesting that decreased Notch signaling may play a role in SCLC development upon *Rnf20* loss.

Importantly, we observed a major upregulation of genes associated with extracellular matrix (ECM) organization, mesenchymal differentiation, as well as the HIF-1 signaling pathway, which was not dependent on H2Bub1 or H3K4me3. HIF1α activation plays a crucial role in the adaptation of cancer cells to hypoxia by inducing the expression of genes involved in angiogenesis, metabolism, and cell survival[56] through its function in promoting Pol II pause release[43]. In this context, *Rnf20* haploinsufficiency resulted in a major decrease in promoter-proximal Pol II pausing and increased expression, particularly of HIF1α targets, which was not dependent on H2Bub1 but was coupled to increased HIF1α protein levels. Consequently, *Hif1a* silencing increased the reduced pausing index in RNF20-haploinsufficient cells to normal levels. Recent studies using double knockdown experiments in breast cancer cells have shown that RNF20-RNF40-mediated H2Bub1, along with changes in H3K4me3, is required for the activation of HIF1 target genes under hypoxia[40]. In our study, we found that, unlike RNF20 loss, RNF40 depletion led to a significant decrease in a subset of HIF-1 target genes. In contrast to RNF20 loss, which markedly promoted cell growth and migration, RNF40 depletion had minimal impact, suggesting distinct

roles for these proteins in lung epithelial and lung cancer cells. This distinction is particularly significant given the opposing expression patterns of RNF20 and RNF40 and their contrasting associations with survival outcomes in AD patients. This contrasts with the roles of these proteins in breast cancer cells, where both RNF20 and RNF40 promote carcinogenesis[57–59]. These differences may be due to their interactions with cell-type-specific binding partners.

HIF1α-mediated metabolic shift, also known as the Warburg effect, allows cancer cells to produce energy via glycolysis even in the presence of oxygen and fully functioning mitochondria and maintain their aggressive phenotype[56,60]. Interestingly, RNF20 depletion in liver cancer cells also resulted in a Warburg effect, suggesting that the identified mechanism could be relevant to a broader range of cancer types[61]. Importantly, silencing of HIF1α or inhibition of glucose uptake rescued the increased growth and migratory ability of *Rnf20 +/-* lung epithelial cells, as well as *Rnf20 +/-* AD and SCLC cells in both cell culture and in vivo experiments. Thus, increased DNA damage, HIF1α activation, together with insufficient p53 and Rb function, might explain the high incidence of SCLC-like lesions upon RNF20 loss. Interestingly, the DNA damage in *Rnf20 +/-* cells and tumors derived from *Rnf20 +/-* cells was also significantly reduced upon *Hif1a* downregulation. Increased γH2AX during hypoxia maintains HIF1α stability and nuclear accumulation, thereby facilitating HIF-1α/hypoxia signaling activation[62], suggesting a feedback mechanism between RNF20 loss, HIF1α activation, and γH2AX accumulation. Moreover, we found a significant negative correlation between RNF20 levels and the levels of HIF1α, the glucose transporter GLUT1 and glycolytic enzymes in lung adenocarcinoma and SCLC patients, suggesting the inhibition of glucose uptake or HIF1α as a promising approach for the treatment of lung cancer patients with decreased *RNF20* levels.

## Methods

### Mouse lines and animal experiments
All animal experiments were performed according to the institutional guidelines and are covered in an approved animal experimental protocol by the Committee for Animal Rights Protection of the State of Baden-Württemberg (Regierungspraesidium Karlsruhe, Experimental protocol Az.: 35-9185.81/G-260/17 and Az.: 35-9185.81/G-119/23). Mice were grouped into cohorts, irrespective of their sex. The *Rnf20tm1a(EUCOMM)Wtsi* line was generated by microinjection of *Rnf20 tm1a(EUCOMM)Wtsi* mouse embryonic stem cells, obtained from the European Conditional Mouse Mutagenesis Program (EUCOMM), into blastocysts[33] (Supplementary Fig. 1a). Animal experiments with this mouse model were performed according to the Experimental protocol Az.: 35-9185.81/G-260/17. Mice were sacrificed at the ages of 6 months and 1 year. Lungs were isolated, fixed with 3.7% formaldehyde, and embedded in paraffin. A total of 7μm μm-thick paraffin sections were used for hematoxylin and eosin staining and IHC staining.

The C57BL/6 J and the BALB/c Nude mouse lines were purchased from Janvier Lab and kept under pathogen-free conditions at the Core

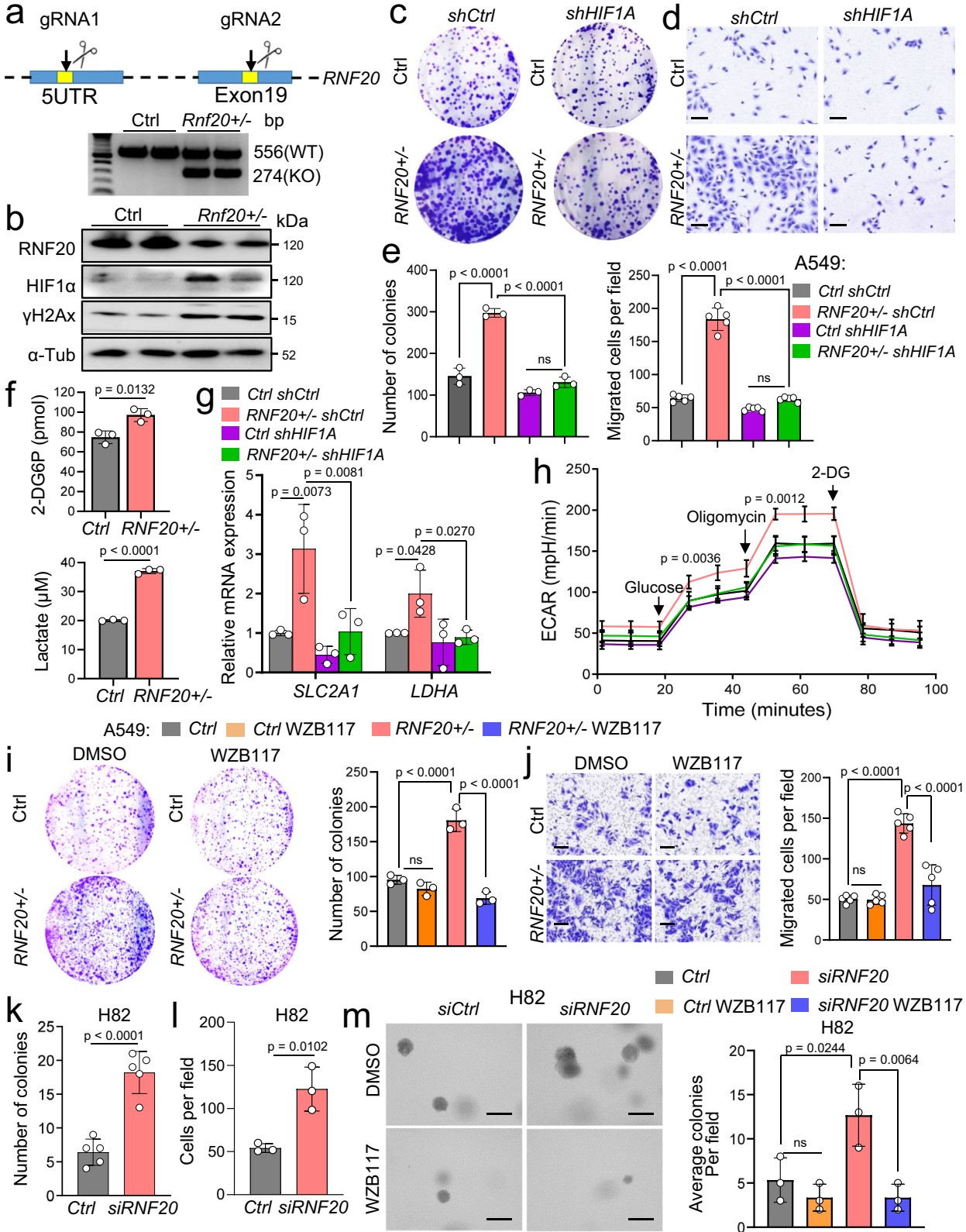

Facility Preclinical Models of Medical Faculty Mannheim. Animal experiments involving subcutaneous or intravenous injections were conducted in accordance with the Experimental Protocol Az.: 35-9185.81/G-119/23.

For induction of experimental metastasis, $1 \times 10^6$ control or *Rnf20 +/-* LLC1 cells or $1 \times 10^7$ control or *RNF20 +/-* A549 cells expressing control shRNA or shRNA against *Hif1a* were injected intravenously into 10–11 weeks old C57BL/6 and BALB/c nu/nu mice, respectively. For subcutaneous transplantation, $5 \times 10^6$ control or *RNF20 +/-* A549 cells were injected into the flanks of 10–11 weeks old BALB/c nu/nu mice. At the experimental endpoint, mice were sacrificed based on a predefined humane endpoint scoring system. The score sheet assessed multiple criteria, including general appearance, body weight loss, natural behavior, tumor burden, and facial expressions, with each category

**Fig. 6 | HIF1α activation upon RNF20 loss promotes cell growth and migration in human adenocarcinoma and small cell lung cancer cells. a** Schematic representation of the strategy used to generate *RNF20 +/-* A549 cells and PCR genotyping results showing distinct bands for WT and *RNF20* KO alleles in control and *RNF20 +/-* A549 cells. The scissors icon is created in BioRender[67]. **b** Western blot analysis for RNF20, HIF1α, γH2AX and α-Tubulin of total cell lysates of control (*RNF20 +/+*) and *RNF20 +/-* A549 cells. **c–e** Clonogenic (*n* = 3) (**c**) and Boyden chamber migration assay (*n* = 5) (**d**) with control and *RNF20 +/-* A549 cells infected either with control shRNA or shRNA against *HIF1a* and quantification (**e**). **f** 2-DG uptake (top) and concentration of lactate in the supernatant (bottom) of control and *RNF20 +/-* A549 cells stably expressing control shRNA or shRNA against *HIF1A* (*n* = 3). **g, h** qPCR analysis of the glucose transporter *SLC2A1 (GLUT1)* and *LDHA* (**g**) and ECAR (**h**) in control and *RNF20 +/-* A549 cells, stably expressing either control shRNA or shRNA against *HIF1A* (*n* = 3). **i** Clonogenic assay with control and

*RNF20 +/-* A549 cells treated either with DMSO or with WZB117 (left) and quantification (right) (*n* = 3). **j** Boyden chamber migration assay with control and *RNF20 +/-* A549 cells treated either with DMSO or with WZB117 (left) and quantification of the number of colonies (right) (*n* = 5). Scale bars, 150μm. **k, l** Quantification of the number of colonies in colony formation assay (*n* = 5) (**k**) and migrated cells per field in a Boyden chamber migration assay (*n* = 3) (**l**) with H82 cells transfected with control siRNA or siRNA against *RNF20*. **m** Soft agar colony-forming assay with H82 cells transfected with control siRNA or siRNA againsit *RNF20* treated with DMSO or with WZB117 (*n* = 3). Statistical analysis in (**f, k, l**) was performed using a two-tailed Student's t-test. Multiple comparisons were performed using one-way ANOVA with Tukey's (**e, h, i, j, m**) or Šídák's (**g**) multiple comparisons test. Data are shown as mean ± SEM; ns, non-significant. 'n' indicates biological replicates.

scored from 0 to 3. Mice were euthanized upon reaching a score of 3 in any single category or a cumulative score of 12. Lungs were isolated, fixed with 3.7% formaldehyde, and embedded in paraffin. 7 μm-thick paraffin sections were used for hematoxylin and eosin staining and IHC staining.

Tumor size was measured and calculated using formula volume = $(L \times W^2)/2$(L length ; W width).

Metastatic area was defined as the percentage of lung area occupied by metastatic tumor, measured by ImageJ. Tumor volumes and number of tumor nodules in the lung were quantitated from H&E stained lung sections using ImageJ. The study compiled with the Experimental protocol Az.: 35-9185.81/G-119/23, which permits a burden of a single tumor with a maximum diameter 1.5 cm or multiple tumors with a maximum total combined diameter 3 cm.

## Cell culture, knockout cell line generation using CRISPR/Cas9-mediated gene editing

MLE12, LLC1, H82, BEAS-2B (B2B), A549, H69, A427, and HEK293T cells were purchased from American Type Culture Collection. MLE12 cells were cultured in DMEM/F12 medium, supplemented with 10% fetal bovine serum (FBS) and 1% penicillin/streptomycin/glutamine (PSG). LLC1 and HEK293T cells were cultured in DMEM medium supplemented with 10% FBS and 1% PSG. H82, H69, A42,7 and A549 cells were cultured in RPMI (Roswell Park Memorial Institute) 1640 medium supplemented with 10% FBS and 1% PSG. For the generation of *Rnf20 +/-* MLE12 and LLC1 cells by the CRISPR/Cas9 system, two guide RNAs (gRNAs): gRNA1-Rnf20 forward Mus Musculus (5'- CACCGT-GAAAAGCTGGAGCGACGCC −3'); gRNA1-Rnf20 reverse Mus Musculus (5'- AAACGGCGTCGCTCCAGCTTTTCAC −3') and gRNA2-Rnf20 forward Mus Musculus (5'- CACCGGATTTCCATCGCATCTACAT −3'); gRNA2-Rnf20 reverse Mus Musculus (5'- AAACATGTAGATGCG ATGGAAATCC −3') were used. The gRNAs were ligated into PX459 V2.0 plasmids. Both plasmids were transfected into MLE12 or LLC1 cells by Lipofectamine 2000 according to the manufacturer's instructions. Transfected cells were selected using puromycin (2 μg/ml) for 48 h and then expanded for single clone selection.

For the generation of a HIF1α knockdown stable cell line, $0.5 \times 10^6$ HEK293T cells were seeded on a 6-well plate and transfected with 1.5 μg plasmids containing shRNA for *Hif1a* (TRCN0000054448) or control shRNA obtained from the RNAi consortium (TRC) shRNA library, along with packaging plasmids using X-tremeGENE DNA transfection reagent (Roche, 6366236001). Viral supernatant was collected 48 h after transfection and used to transduce control and *Rnf20 +/-* MLE12 in the presence of 0.1% polybrene. 48 h after transduction, cells were selected with 5 μg/ml puromycin for two passages and were maintained in a normal medium containing 2 μg/ml puromycin.

For the generation of *RNF20 +/-* A549 cells, two guide RNAs (gRNAs): gRNA1-RNF20 forward Homo sapiens (5'- CACCGTCAGAC GGCCGATTGGCTGACGG −3'); gRNA1-RNF20 reverse Homo sapiens

(5'- AAACTCAGCCAATCGGCCGTCTGAC −3') and gRNA2-RNF20 forward Homo sapiens (5'- CACCGGGAGGGCACTACCACTACGCAGG −3'); gRNA2-RNF20 reverse Homo sapiens (5'- AAACGCGTAGTGGTAGT GCCCTCCC −3'), were used. The gRNAs were ligated into lenticrispr v2 plasmids, and both plasmids were transfected into A549 cells by Lipofectamine 2000 according to the manufacturer's instructions. Transfected cells were selected using puromycin (4 μg/ml) for 48 h and then expanded for single clone selection, followed by genotyping and Sanger sequencing.

## siRNA transfection

Human *RNF20* siRNA (ON-TARGET plus siRNA, SMART Pool, L-007027-00-0005) and control siRNA were purchased from Horizon Discovery. $10^5$ H82 cells were seeded in a 6-well plate and transfected with 25 nM siRNAs using Lipofectamine RNAiMax (Thermo Fisher Scientific, 13778-075). After overnight incubation with the transfection mix, fresh medium was added and the cells were harvested for RNA and protein isolation 72 h after transfection or used for further assays.

## Chemical treatments

For DNA damage response assays, $H_2O_2$ was added to a final concentration of 0.5 mM for 1 h. Cells were harvested 4 h after treatment for total protein isolation. For immunofluorescence staining of γH2AX, cells were washed with PBS and fixed with 4% PFA at 4 h, 8 h, 16 h, and 24 h after treatment.

For hypoxia treatment, cells were cultured under 1% $O_2$, 94% $N_2$ and 5% $CO_2$ at 37 °C for 12 h.

WZB117, a glucose transporter 1 inhibitor, was purchased from Selleck Chemicals (S7927). WZB117 was added at a final concentration of 10 μM for 24 h, followed by Boyden chamber migration assay and Seahorse assays in fresh medium. For colony formation assays, cells were cultured in the continued presence of WZB117.

## Histology, immunohistochemistry (IHC), immunofluorescence (IF) staining

For hematoxylin and eosin (HE) staining, slides were dewaxed by heating at 55 °C for 10 min, incubation with xylene for 3×5 min, followed by rehydration steps (100% ethanol, 75% ethanol, 50% ethanol, PBS, 5 min each). Lung sections (7μm thick paraffin sections) were stained either with HE or using antibodies. Staining with HE (GHS116, HT-110216; Sigma-Aldrich) was performed according to the manufacturer's instructions. For immunohistochemistry, after rehydration, the slides were heated in 10 mM citrate buffer for 10 min in a microwave for antigen retrieval and cooled at room temperature. The subsequent processes were performed with the VECTASTAIN Universal Quick HRP Kit (PK-7800; Vector Laboratories) according to the manufacturer's instructions. DAB Peroxidase (HRP) Substrate Kit (SK-4100; Vector Laboratories) was used for color development. Images were acquired with an Axio Scan.Z1 slide scanner (ZEISS).

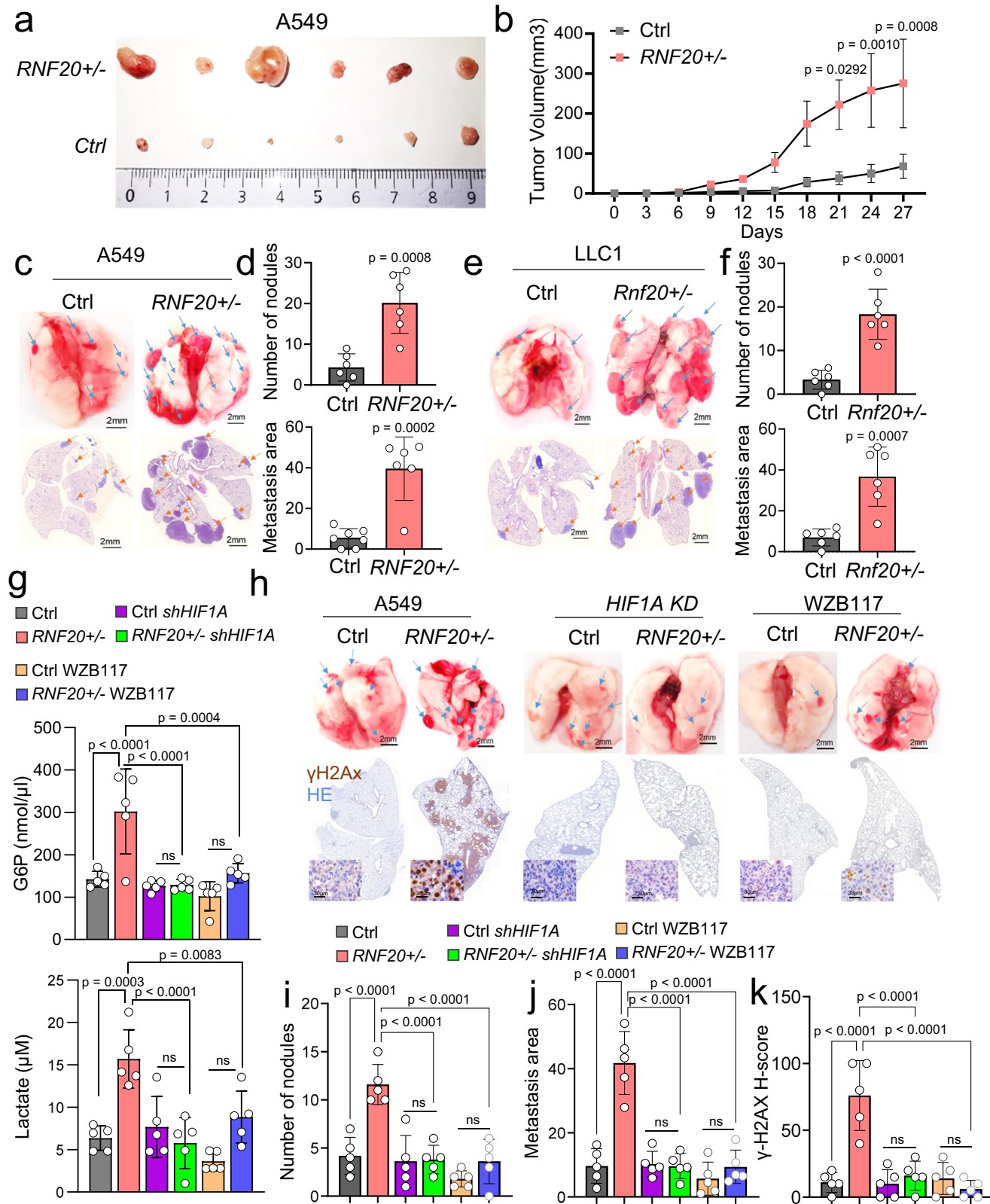

For immunostaining of cells, 100,000 cells were seeded on coverslips in 24-well plates. Next, cells were washed with PBS and fixed with 4% PFA for 10 min. Next, unspecific binding was blocked and cells were permeabilized with 5% BSA, 0.2% Triton X-100 in PBS for 1 h, followed by incubation with primary antibody in blocking buffer overnight at 4 °C. After three washes with PBS, cells were incubated with secondary antibody for 2 h, followed by incubation with DAPI for 20 min. Next, slides were washed and mounted with ProLong Gold antifade mountant (P36930, Life Technologies). The fluorescence images were acquired by using a Zeiss LSM 700 confocal microscope with a 25× or 63× objective and analyzed with ImageJ software. A list of the primary and secondary antibodies used in this study is provided in Supplementary Data 5.

**Fig. 7 | RNF20 loss promotes tumor growth and metastasis through HIF1α activation and metabolic rewiring. a** Representative tumors harvested on day 27 after subcutaneous injection of control and *RNF20 +/-* A549 cells into the flanks of BALB/C Nude mice (*n* = 6). **b** Quantification of the macroscopic tumor volume at different days after injection. **c** Control and *RNF20 +/-* A549 cells were injected intravenously into the tail vein of BALB/C Nude mice (*n* = 6). Macroscopic appearance (upper panels) and HE staining (lower panels) of representative lungs. Scale bars, 2 mm. **d** Quantification of the metastatic nodules (upper panels) and the metastasis area (lower panels), in lung sections of mice injected with control and *RNF20 +/-* A549 cells. **e** Control and *Rnf20 +/-* LLC1 cells were injected intravenously into the tail vein of C57BL/6 mice (*n* = 6). Macroscopic appearance (upper panels) and HE staining (lower panels) of representative lungs. Scale bars, 2 mm. **f** Quantification of the metastatic nodules (upper panels) and the metastasis area (lower panels), in lung sections of mice injected with control and *Rnf20 +/-* LLC1 cells. **g** Concentration of glucose-6-phosphate (G6P, top panel) and lactate in

lung homogenates (bottom panel) of BALB/C Nude mice injected intravenously into the tail vein with control and *RNF20 +/-* A549 cells, as well as control and *RNF20 +/-* A549 cells after shRNA mediated *HIF1A* silencing or after treatment with the GLUT1 inhibitor WZB117 (*n* = 5). **h** Macroscopic appearance (upper panels) and γH2AX combined with HE staining (lower panels) of representative lungs. Scale bars, 2 mm. **i–k** Quantification of the metastatic nodules (**i**), metastasis area (**j**) in lung sections and the relative staining intensity (H-Score) for γH2AX (**k**) in lung tumors of BALB/C Nude mice injected with control and *RNF20 +/-* A549 cells, as well as control and *RNF20 +/-* A549 cells after shRNA mediated *HIF1A* silencing or after treatment with the GLUT1 inhibitor WZB117 (*n* = 5). Statistical analysis in (**d, f**) was performed using a two-tailed Student's t-test. Multiple comparisons were performed using two-way RM ANOVA with Šídák's multiple comparisons test (**b**) or using one-way ANOVA with Tukey's multiple comparisons test (**g, i, j, k**). Data are shown as mean ± SEM. ns, non-significant. 'n' indicates biological replicates.

## Human tissue microarray quantification of immunoreactivity

Human lung cancer tissue microarray slides, containing 168 cases of multiple types of lung cancer, as well as 10 normal and 10 cancer adjacent normal tissue samples, were purchased from US Biomax lnc (LC2085c). IHC staining was performed as described above. For semiquantitative analysis of immunoreactivity of RNF20, HIF1α, LDHA and GLUT1, H-score of the nuclear staining signal for RNF20 and HIF1α and the staining signal for LDHA and GLUT1 of cells was quantified as follows: ten fields were randomly selected with at least 100 cells in each field, and the H-score was generated by adding the percentage of strongly stained (×3), the percentage of moderately stained (×2), the weakly stained (×1) cells, giving a range of 0–300. The score was obtained by two of the authors independently. The differences were less than 5%, and the mean of the two values was used. At least 1000 cells in 10 random fields were analyzed.

## MitoSOX Red staining and total ROS assay

Mitochondrial ROS was measured using MitoSOX Red Mitochondrial Superoxide Indicator (Thermo Fischer, M36008), according to the manufacturer's instructions. Briefly, $10^5$ cells were seeded on coverslips in a 24-well plate and washed three times with PBS before adding 5 µM MitoSOX reagent for 10 min at 37 °C. After three washes with PBS, cells were stained with DAPI for 20 min, and then mounted with Mowiol 4–88 (Millipore, 475904) mounting medium for imaging. For total ROS, $10^5$ cells were seeded on coverslips in a 24-well plate and washed three times with PBS. The total ROS was measured using Total Reactive Oxygen Species (ROS) Assay Kit 520 nm (Thermo Fischer, 88–5930), according to the manufacturer's instructions. Cells were mounted with Mowiol for imaging. In both assays, cells treated with 200 µM $H_2O_2$ were used as a positive control.

## Western blotting

Cells were cultured in a normal medium until reaching 70%-80% confluence, after which they were washed with PBS and lysed in radio-immunoprecipitation assay (RIPA; 50 mM Tris-HCl, pH 8.0; 150 mM NaCl; 1% NP-40; 0.5% sodium deoxycholate; 0.1% SDS) buffer supplemented with proteinase and phosphatase inhibitors. The lysate was harvested by scraping. The concentration of protein was measured by the Pierce BCA protein assay kit (Pierce Biotechnology, 23225). The same amount of protein was mixed with 5x Leammli sample buffer and boiled at 95 °C for 10 min. Protein extracts were loaded on the SDS-PAGE gel, separated by electrophoresis, and then transferred onto nitrocellulose membranes (Sartorius, 11306------41BL). Membranes were stained by Ponceau for 1 min and then blocked with blocking buffer (PBS + 0.1%Tween-20 + 5%BSA) at room temperature for 1 h. Afterwards, the membrane was incubated with primary antibody at 4 °C overnight, followed by incubation with appropriate secondary

antibodies at room temperature for 2 h. Immunoreactive bands were detected by chemiluminescence. Three independent experiments were performed.

## Scratch wound and Boyden chamber migration assays

Control and *Rnf20 +/-* MLE12, LLC1 and A549 cells were seeded in a 6-well plate. When cells reached 100% confluence, scratches were made by 10 µl pipette tips, and the cells were cultured in medium without FBS for 24 h and 48 h. For Boyden chamber assays, inserts (Corning, 353097) were placed on a 24-well plate that contained normal culture medium. $10^5$ MLE12, $5 × 10^4$ LLC1 and $5 × 10^4$ A549 cells were seeded on the top of the inserts in medium without FBS. After 6 h, the insert membranes were fixed with 4% PFA for 10 min and washed 3 times with PBS. The upper surface of the membranes was cleaned by wiping with a cotton swab followed by staining with crystal violet for 10 min. After three washes with PBS, the membranes were cut and mounted on slides. Images were acquired from a random area of the membrane with a 20× objective, and the cells which migrated to the lower surface of the membrane were quantified using ImageJ. For each migration assay, a minimum of three independent experiments were performed.

## Soft agar and plate colony formation assay

For plate colony formation assays, 3000 MLE12 and 1000 LLC1 and A549 cells were seeded in 6-well plates, after 10 days, the colonies were stained with Crystal Violet (Thermo Fisher). For soft agar assays, 5000 cells were suspended in the complete medium containing 0.3% low-melting agarose (Carl Roth) and plated on solidified 0.6% agarose in complete normal medium in 6-well plates. The colonies were stained with 0.005% Crystal Violet after 14 days.

## Measurement of glucose uptake, lactate secretion and Seahorse assays

Glucose Uptake Colorimetric Assay Kit and Lactate Assay Kit were used to measure glucose uptake and lactate secretion according to the manufacturer's instructions (MAK083-1KT and MAK065-1KT). Glucose-6-phosphate assay kit was used to measure the generation of Glucose-6-phosphate based on the manufacturer's instructions (MAK014-1KT). The extracellular acidification rate (ECAR) was measured using the Seahorse XFe 24 Extracellular Flux Analyzer using the Seahorse XFe Glycolysis Stress Test Kit. Briefly, $5 × 10^4$ MLE12 cells were seeded into a Seahorse XFe 24 cell culture microplate. After calibration and baseline measurements, glucose (10 mM), the oxidative phosphorylation inhibitor oligomycin (1 µM), and the glycolytic inhibitor 2-DG (50 mM) were sequentially injected into each well. The data was analyzed by Seahorse XFe 24 Wave software. Four technical replicates for each biological replicate was measured. Three independent experiments were performed.

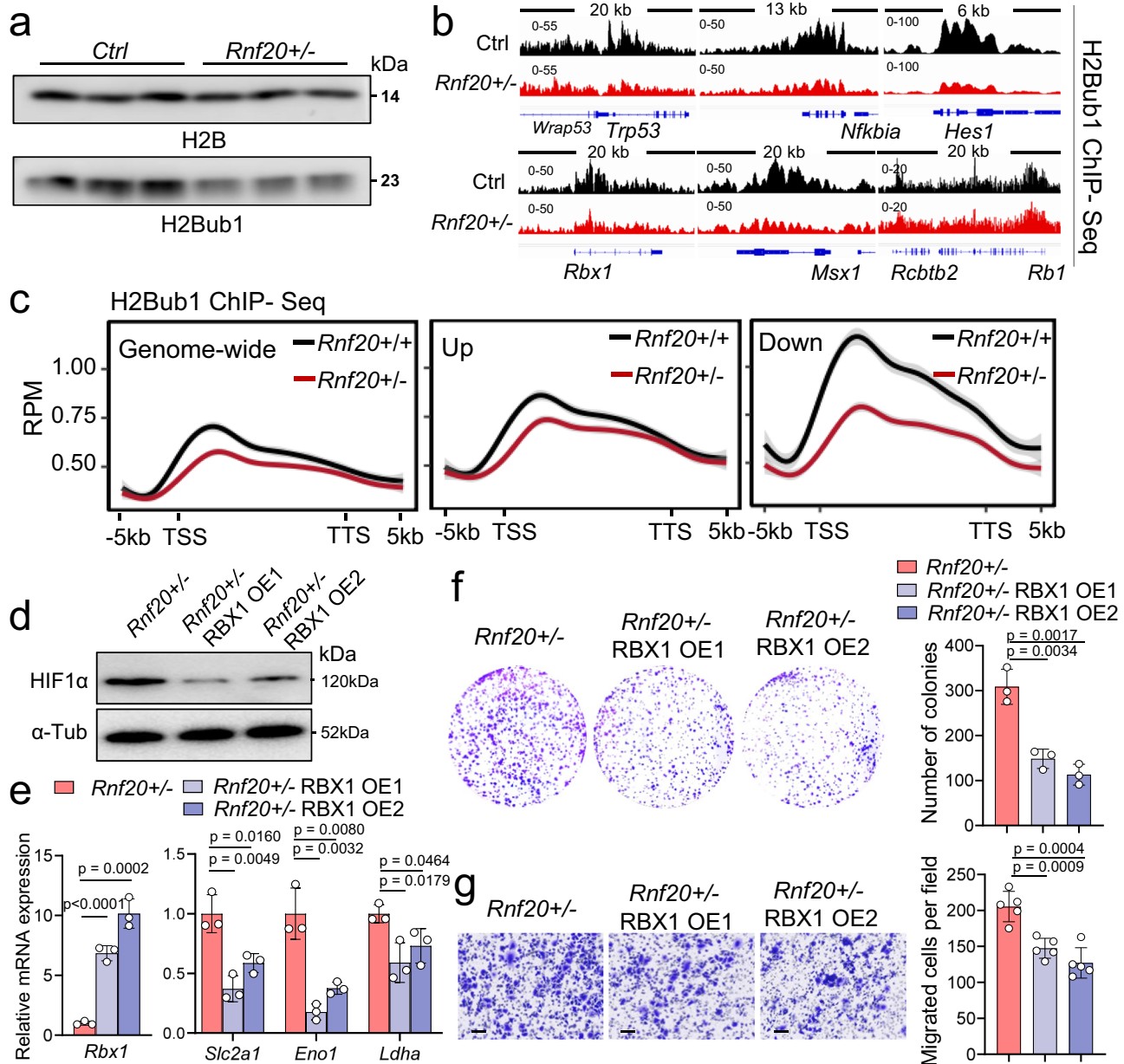

**Fig. 8 | RNF20 controls HIF1α degradation through RBX1. a** Representative western blot analysis of H2Bub1 in control and *Rnf20 + /-* MLE12 cells. **b** Genome tracks of merged H2Bub1 ChIP-Seq reads in control and *Rnf20 + /-* MLE12 cells at genes downregulated upon *Rnf20* LOF. H2Bub1 levels are majorly decreased at *p53 (Trp53)*, *Nfkbia*, *Hes1*, *Rbx1* and *Msx1* genes, while no change was observed at *Rb1*. **c** Average genome-wide H2Bub1 ChIP-Seq signal as well as H2Bub1 ChIP-Seq signal at genes upregulated and downregulated upon *Rnf20* loss in control and *Rnf20 + /-* MLE12 cells. Line plots represent mean values, with shaded areas representing the ± SEM across *n* = 2 biological replicates for each group. RPM, Reads Per Million mapped reads. **d** Western blot analysis of HIF1α in total cell lysates from

*Rnf20 + /-* MLE12 cells or *Rnf20 + /-* stably expressing RBX1 (OE1 and OE2 stand for two stable pools). **e** Relative mRNA expression of *Rbx1*, *Slc2a1*, *Eno1* and *Ldha* in *Rnf20 + /-* MLE12 cells (*n* = 3) or *Rnf20 + /-* stably expressing RBX1 (*n* = 3). **f** Clonogenic assay with *Rnf20 + /-* MLE12 cells or *Rnf20 + /-* stably expressing RBX1 (left panels) and quantification of the number of colonies (right panel, *n* = 3). **g** Boyden chamber migration assay with *Rnf20 + /-* MLE12 cells or *Rnf20 + /-* stably expressing RBX1 (left panels) and quantification of the number of migrated cells per field (right panel, *n* = 5). Statistical analysis in (**e, f, g**) was performed using a two-tailed Student's t-test. Data are shown as mean ± SEM. 'n' indicates biological replicates.

## LC-MS/MS (Liquid Chromatography-Tandem Mass Spectrometry) Data Acquisition and Analysis

Metabolites of the glycolysis, pentose phosphate pathway, and TCA cycle were analyzed as previously described[63]. In brief, 1 million cells were used and metabolites were extracted with ice-cold methanol/water (85/15, v/v). Isotope-labeled internal standards were added, and samples were evaporated in a vacuum concentrator (Eppendorf, Hamburg, Germany) at 30 °C. The samples were resolved in 50 μL methanol/water (50/50, v/v) and subsequently transferred to the LC-MS/MS system. Liquid

chromatography was performed on an Agilent 1290 Infinity pump system (Agilent, Waldbronn, Germany) with a Phenomenex Luna Amino-column (100 mm × 2.0 mm, 3 μm) with ammonium acetate (10 mmol/L, pH 9.0) as mobile phase A and 100 % acetonitrile as mobile phase B. Five μL per sample were injected. The column temperature was set at 30 °C. The gradient with a flow rate of 700 μL/ minute was as follows: 0–1 min, 5 % A; 1–3 min, 5–60% A; 3–15 min, 60–95 % A; 15–18 min, 95 % A; 18–18.1 min, 95–5 % A; 18.1–24.1 min, 5 % A. Mass spectrometry was performed using a QTrap 5500 mass spectrometer (Sciex, Darmstadt, Germany) with

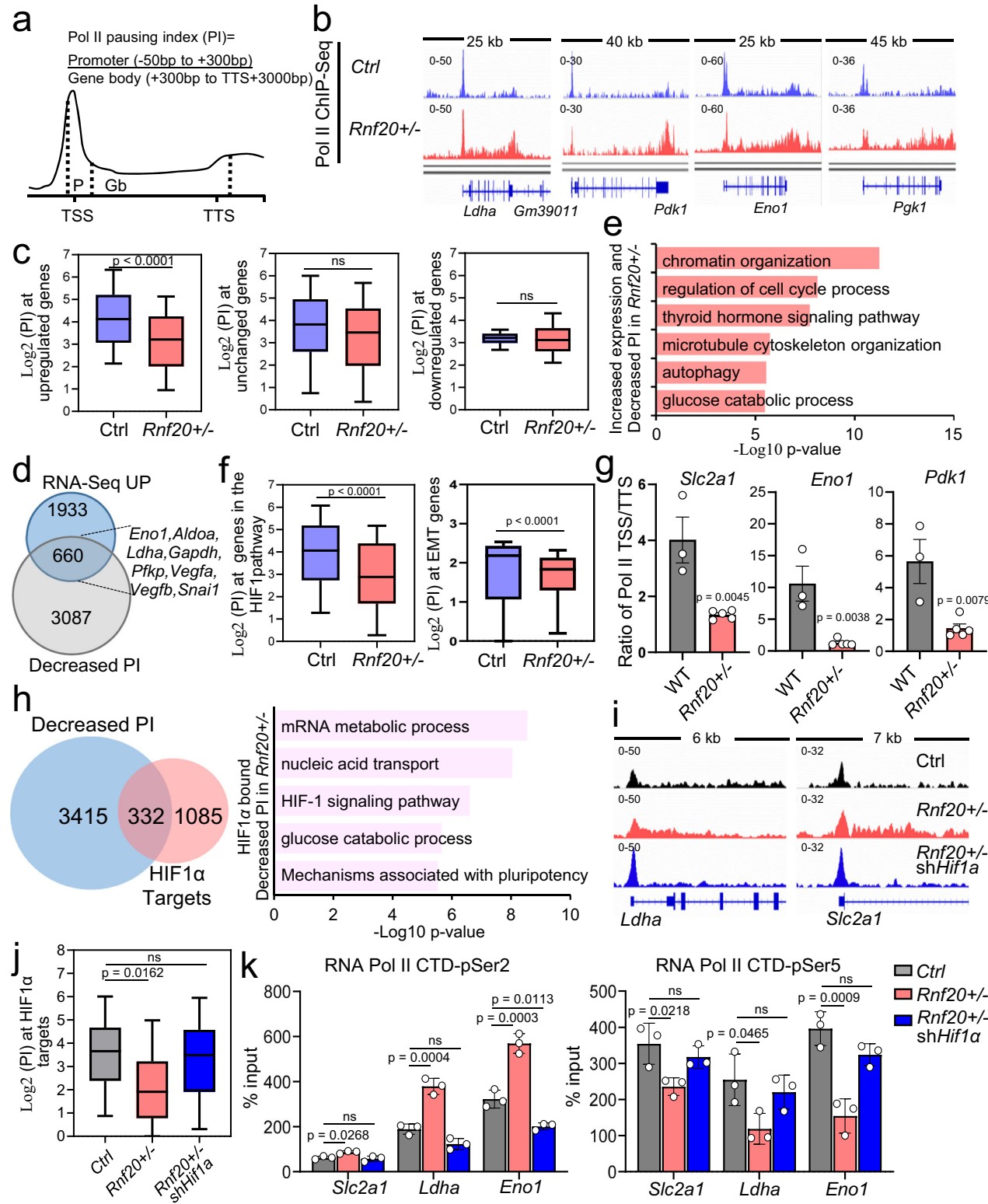

electrospray ionization in negative mode. ESI parameters were set to TEM 400 °C, IS −4500 V, CUR 25 psi, GS1 40 psi, and GS2 60 psi. Data acquisition and instrument control were managed using Analyst 1.6.2 software. Peak integration, data processing, and quantification were performed using MultiQuant 3.0 (both Sciex, Darmstadt, Germany). Area under the peak was used for quantification of the metabolites, and the specific MRM transitions were normalized to the appropriate isotope-labeled internal standards and to the protein content of the sample.

## ChIP-sequencing and ChIP-qPCR analysis

$10^7$ control, *Rnf20 + /-*, and *Rnf20 + /-*-HIF1α KD MLE12 cells were fixed with 1% methanol-free formaldehyde for 10 min at room temperature. 125 mM glycine was added to quench the fixation. After three washes with PBS, cells were suspended in cold PBS at a concentration of $5 \times 10^6$/ml. An equal volume of 2× lysis buffer (100 mM Tris-HCl pH8, 300 mM NaCl, 2% Triton X-100, 10 mM CaCl$_2$) was added for 10 min on ice with gentle shaking. The nuclei were washed with cold PBS and centrifuged at 2500 *g*, 4 °C for 5 min.

**Fig. 9 | HIF1α activation upon *Rnf20* haploinsufficiency induces RNA polymerase II promoter-proximal pause release at metabolic genes. a** Schematic representation of the method for calculating Pol II pausing index (PI). **b** Genome tracks of merged Pol II RNA-Seq reads of control and *Rnf20 +/-* MLE12 cells at genes involved in glycolysis. **c** Log2 of the PI at genes upregulated, unchanged or downregulated upon *Rnf20* LOF ($n = 3$). **d, e** Venn diagram showing an overlap of genes with decreased PI and upregulation in *Rnf20 +/-* compared to control MLE12 cells (**d**) and GO analysis of the overlapping genes (**e**). **f** Log2 of the PI at HIF-1 signaling pathway (left) or EMT genes (right) in control and *Rnf20 +/-* MLE12 cells ($n = 3$). **g** Ratio of the Pol II enrichment at the TSS and TTS of *Scl2a1*, *Eno1* and *Pdk1* in control ($n = 3$) and *Rnf20 +/-* lungs ($n = 5$) determined by Pol II Chip-qPCR. TSS, Transcription Start Site; TTS, Transcription Termination Site. **h** Venn diagram showing the overlap of genes with decreased PI in *Rnf20 +/-* compared to control

MLE12 cells and HIF1α-bound genes in A549 lung AD cells[44] (left) and GO analysis of the overlap (right). **i, j** Genome tracks of merged Pol II ChIP-Seq reads of control, *Rnf20 +/-* and *Rnf20 +/-* MLE12 cells stably expressing shRNA against *Hif1a* (**i**) and PI at HIF1α-bound genes in lung A549 AD cells (**j**, $n = 2$). **k** Chip-qPCR for RNA Pol II CTD-pSer2 (left) and CTD-pSer5 (right) at *Scl2a1*, *Ldha* and *Eno1* gene body in control, *Rnf20 +/-* MLE12 cells, or *Rnf20 +/-* MLE12 cells after shRNA mediated *Hif1a* silencing ($n = 3$). Statistical analysis in (**c, f, g**) was performed using a two-tailed Student's t-test. Multiple comparisons were performed using one-way ANOVA with Šídák's (**j, k**) multiple comparisons test. Boxplots in (**c, f, j**) represent the minimum, 25th percentile (Q1), median, 75th percentile (Q3), and maximum of the Log2 PI in control, *Rnf20 +/-* MLE12 cells, or *Rnf20 +/-* MLE12 cells after shRNA-mediated *Hif1a* silencing. GO analysis in **e** and **h** was performed using Metascape (Version 3.5). Data are shown as mean ± SEM. ns, non-significant. 'n' indicates biological replicates.

Chromatin was sheared using a Covaris Ultrasonicator (10% duty factor, 200 cycles, 75 A) for 6 min and the samples were clarified at $16,000 g$ for 15 min. Next, the extracts were pre-cleaned with 75 µl protein G beads for 2 h at 4 °C, then incubated with 0.5 µg Pol II antibody (14958S, Cell Signaling), 1 µg Pol II p-Ser5 (ab5131, Abcam), 1 µg Pol II p-Ser2 (13499 s, Cell Signaling), 1 µg H2Bub1 (5546S, Cell Signaling) or 1 µg H3K4me3 (ab8580, Abcam) antibodies overnight at 4 °C, followed by binding to 75 µl BSA-coated protein G beads. Immunoprecipitates were washed twice with low salt buffer (0.1% SDS, 1% Triton X-100, 2 mM EDTA, 20 mM Tris-HCl pH7.5, 150 mM NaCl), three times with high salt buffer (0.1% SDS, 1% Triton X-100, 2 mM EDTA, 20 mM Tris-HCl pH7.5, 500 mM NaCl), twice with LiCl buffer (10 mM Tris-HCl pH7.5, 250 mM LiCl, 1% NP-40, 1% sodium deoxycholate, 20 mM EDTA) and then eluted with Elution buffer (50 mM Tris-HCl pH8.0, 10 mM EDTA pH8.0, 1% SDS). Samples were treated with RNase A for 1 h at 37 °C, followed by Protein K digestion for 2 h at 37 °C, and cross-linking was reversed by incubation at 65 °C overnight. DNA was purified by using the Qiagen mini elute PCR purification kit (Qiagen, 28004).

Libraries were prepared with the NEBNext Ultra II DNA Library Prep Kit for Illumina (NEB, E7645S/L, E7103S/L) according to the manufacturer's instructions.

ChIP-Seq reads were trimmed using Trimmomatic, at a minimum length of 60 bp, and a quality score of a minimum of 15. Trimmed reads were mapped to mouse genome mm10 (UCSC assembly) with the help of bowtie2 (default settings). PCR artifacts were removed using the MarkDuplicates.jar function from Picard 1.136. Peaks were called using MACS14[64] (default settings). Peaks overlapping the blacklist defined by ENCODE were removed. Genome-wide distribution of the reads was performed using plotProfile from the deepTools suite[65]. Individual bam-mapped files were merged with the help of bamtools merge (default settings). Merged bam files were converted to BigWig format using BamCoverage from deepTools (-b 20 -smooth 40 --normalizeUsing RPKM -e 150). The pausing index was calculated by dividing the normalized count per million reads (CPM) on the TSS area (−50 to 300 bp) by the CPM on the gene body plus 3 kb after the transcription termination site (TTS). For the calculation, the GitHub repository code (https://github.com/MiMiroot/PIC) was used with settings mm10.gtf --TSSup 50 --TSSdown 300 --GBdown 3000) and ENSEMBL mm10, version 108.

The t-test from the package rstatix was used to calculate the differential PI values from total Pol II ChIP-Seq of *Rnf20 +/-* versus control MLE12 cells. Genes as deceased PI were defined as those Log2 FC ≤ −0.58 with $p < 0.05$.

For ChIP-qPCR, 0.1 ng purified DNA from each sample was used. qPCR was performed by using the SYBR Green PCR master mix (A25742, Applied Biosystems) or qPCRBIO SyGreen Blue Hi-ROX (Nippon Genetics).

## RNA isolation and qPCR
RNA was isolated by using TRIzol RNA Isolating Reagent (Invitrogen, 15596018). For quantitative PCR (qPCR) analysis, cDNA was synthesized with the High-Capacity cDNA Reverse Transcription Kit (Cat # 4368813, Applied Biosystems), and qPCR was performed by using the SYBR Green PCR master mix (A25742, Applied Biosystems) or qPCRBIO SyGreen Blue Hi-ROX (Nippon Genetics). Cycle numbers were normalized to these of β-actin. A list of primers used in this study is provided in Supplementary Data 6.

RNA for RNA-Sequencing was isolated by using the RNeasy microarray kit (Qiagen, 73304).

## RNA-Seq and data analysis
Total RNA was extracted from control and Rnf20 +/- MLE12 cells using the RNeasy Universal Mini Kit (Qiagen, Cat. No. 73404). RNA integrity was assessed with the Agilent 2100 Bioanalyzer using the RNA 6000 Nano Kit (Agilent, Cat. No. 5067-1511). Polyadenylated poly(A) + RNA was enriched for library preparation, and standard sequencing was performed on the DNBSEQ-50 platform.

Raw reads were trimmed with trimmomatic-0.36. The trimmed reads were mapped to the mm10 reference genome using STAR. Differential expression was quantified and normalized using DESeq2 (Version 1.40.0). Reads per kilobase per million mapped (rpkm) was determined using rpkm.default from EdgeR. Excel was used to filter differentially regulated genes. Genes as upregulated were defined as those with Log2 FC ≥ 0.58 and $p < 0.05$, and Log2 FC ≤ −0.58 and $p < 0.05$ as downregulated from RNAseq. GO analysis was performed using Metascape (Version 3.5)[66], KEGG pathway analysis was performed using DAVID Bioinformatics.

## Statistical analysis
All experiments were performed at least three times in biological replicates, and the respective data were used for statistical analyses. The 'n' numbers indicated in the figure legends always refer to biological replicates. For cell culture analyses involving genetically modified cell lines, 'n' always represents biologically independent clones unless stated otherwise. Differences between groups were assessed using an unpaired two-tailed Student's t-test or ANOVA for multiple comparisons. Statistical significance was indicated as follows: *$P < 0.05$, **$P < 0.01$, ***$P < 0.001$. Barplots and Boxplots were produced with GraphPad Prism v8.0.2 and R4.4.2. Numeric p-values are shown within the figures.

## Reporting summary
Further information on research design is available in the Nature Portfolio Reporting Summary linked to this article.

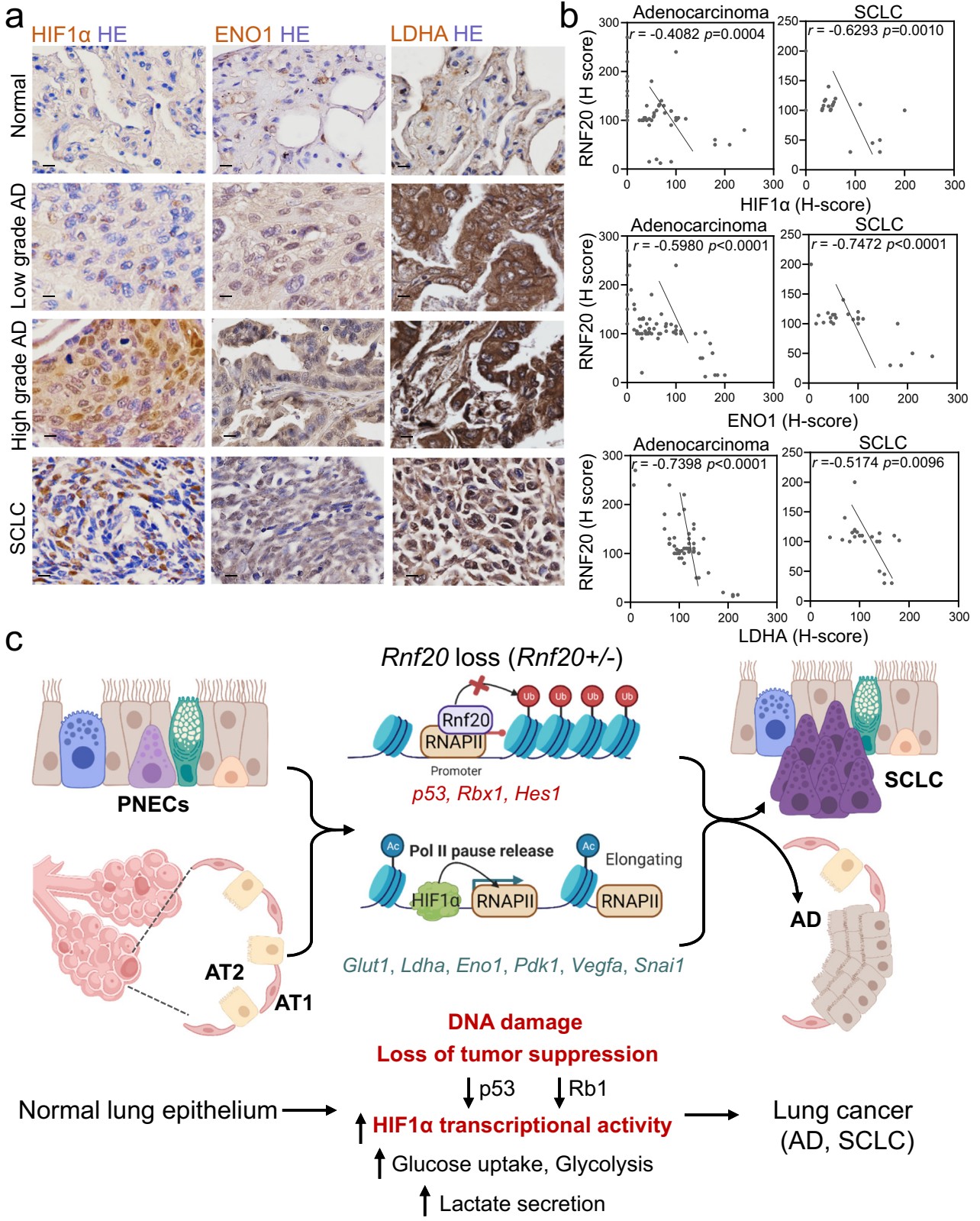

**Fig. 10 | Decreased RNF20 levels correlate with increased levels of HIF1α and glycolytic enzymes in AD and SCLC patients. a** Immunohistochemistry of tissue samples from different types and grades of lung tumors from a tissue microarray stained with HIF1α, ENO1 and LDHA antibodies. Scale bars, 100 μm. **b** Pearson correlation (r) of the relative staining intensity (H-Score) of RNF20 and HIF1α (top panels), RNF20 and ENO1 (middle panels), and RNF20 and LDHA (bottom panels) in AD and SCLC patients, with a two-sided p-value computed using the t-distribution. $n = 70$ for AD; $n = 22$ for SCLC. **c** Schematic model of the function of RNF20 in lung SCLC and AD development. PNEC pulmonary neuroendocrine cell, AT1 alveolar type 1 cell, AT2 alveolar type 2 cell, RNAPII RNA polymerase II. The figure was created in BioRender[68].

## Data availability

The RNA-Seq and ChIP-Seq data generated in this study have been deposited in the Gene Expression Omnibus (GEO) under accession number GSE231317. Processed sequencing data are included in Supplementary Data 1, 3 and 4. Processed metabolomics data are provided in Supplementary Data 2. Source data are provided with this paper.

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

## Acknowledgements

G.D. was supported by the TRR81 (Project B07) and the CRC 1366 (project A03 to GD; project A06 to JH; project C02 to A.C.) funded by the Deutsche Forschungsgemeinschaft (DFG), and the Baden-Württemberg foundation special program "Angioformatics single cell platform". A.C. was supported by the RTG2727 – 445549683 (B1.2) funded by the DFG. The work of M.P. was supported by the European Research Council (ERC) Consolidator Grant EMERGE (no. 773047) and the DFG-funded CRC 1531 (Project-ID 456687919).

## Author contributions

H.L., Y.T., A.S., A.M., J.V., Y.J., R.G., G.P., and S.-I.B. performed the experiments and analyzed the data. H.L., Y.T., B.K.G., S.-I.B., and J.C. performed the bioinformatic analysis. T.A., R.H., M.S., J.H., T.W., G.B., A.C., M.P., and R.S. provided reagents and valuable intellectual input. G.D. designed the experiments, analyzed the data, and wrote the manuscript. All authors discussed the results and commented on the manuscript.

## Funding

## Competing interests

The authors declare no competing interests.

## Additional information

¹Department of Cardiovascular Genomics and Epigenomics, European Center for Angioscience (ECAS), Medical Faculty Mannheim, Heidelberg University, Mannheim, Germany. ²Max Planck Institute for Heart and Lung Research, 61231 Bad Nauheim, Germany. ³Department of Microvascular Biology and Pathobiology, European Center for Angioscience (ECAS) and Mannheim Institute for Innate Immunoscience (MI3), Medical Faculty Mannheim, Heidelberg

University, Mannheim, Germany. [4]Institute of Neuropathology, Justus Liebig University, Giessen, Germany. [5]Centre for Organismal Studies (COS), Heidelberg University, Heidelberg, Germany. [6]Translational Lung Research Center (TLRC) Heidelberg, German Center for Lung Research (DZL), Heidelberg, Germany. [7]Translational Research Unit, Thoraxklinik at Heidelberg University Hospital, Heidelberg, Germany. [8]Cardiovascular Physiology, European Center for Angioscience, Medical Faculty Mannheim, Heidelberg University, Mannheim, Germany. [9]German Centre for Cardiovascular Research (DZHK), Mannheim, Germany. [10]Experimental Pharmacology, European Center for Angioscience, Medical Faculty Mannheim, Heidelberg University, Mannheim, Germany. [11]Université de Lorraine, CNRS, Laboratoire IMoPA, UMR, 7365 Nancy, France. [12]Immunobiochemistry, Mannheim Institute for Innate Immunoscience (MI3) and European Center for Angioscience (ECAS), Medical Faculty Mannheim, Heidelberg University, Mannheim, Germany. [13]DKFZ-Hector Institute, Medical Faculty Mannheim, Heidelberg University, Mannheim, Germany. [14]Angiogenesis & Metabolism Laboratory, Center of Vascular Biomedicine, Berlin Institute of Health at Charité -Universitätsmedizin Berlin, Berlin, Germany. [15]Max Delbrück Center for Molecular Medicine in the Helmholtz Association, Berlin, Germany. [16]Institute for Vascular Signalling, Centre for Molecular Medicine, Goethe University, Frankfurt am Main, Germany. [17]Department of Vascular Dysfunction, ECAS, Medical Faculty Mannheim, Heidelberg University, Mannheim, Germany. [18]Institute for Lung Health (ILH), Justus Liebig University, 35392 Giessen, Germany. [19]Max Planck Institute for Heart and Lung Research, Member of the German Center for Lung Research (DZL), Member of the Cardio-Pulmonary Institute (CPI), Bad Nauheim 61231, Germany. [20]Helmholtz-Institute for Translational AngioCardioScience (HI-TAC) of the Max Delbrück Center for Molecular Medicine in the Helmholtz Association (MDC) at Heidelberg University, 69117 Heidelberg, Germany. [21]These authors contributed equally: Hao Liu, Yongqin Tang, Anshu Singh. ✉e-mail: Gergana.Dobreva@medma.uni-heidelberg.de

