## [Transparent Peer Review file · Nature Communications]

RNF20 links the DNA damage response and metabolic rewiring in lung cancer through HIF1 α

Corresponding Author: Professor Gergana Dobreva

Version 0:

Reviewer comments:

Reviewer #1

(Remarks to the Author)

Liu, Tang, Singh et al. establish a lung cancer genetically-engineered mouse model (GEMM) with Rnf20 haploinsufficiency giving rise to tumors resembling lung adenocarcinoma and small cell carcinoma. Rnf20 inactivation leads to increased DNA damage, HIF1 α upregulation and metabolic reprogramming, and to increased EMT and migratory phenotype in vitro, potentially as a consequence of HIF1 α effect at pausing transcription of genes involved in metabolism and EMT, among others.

Overall, even if the manuscript is well-written and the experiments well designed, I am not sure that the right models were used to support conclusions, and the clinical implications are unclear. There is a number of concerns:

Major comments

- Figures 2A and 2B are not really supportive of authors conclusions, as in each of these, only one cell line versus another cell line is compared. Authors should provide data on additional datasets containing mRNA data of normal lung and LUAD. Also, if the authors want to make any claim regarding the potential of RNF20 downregulation as driver of LUADs, they should comment on the dispersion of RNF20 expression in LUAD, suggestive that that might be the case only in a subset of LUADs.

- Regarding the clinical impact of the findings, by the data shown by the authors it seems that RNF20 downregulation may only have a clinically relevant effect in LUAD (Figure 1G) but not in other lung tumor types, including SCLC (Figure S1F). Thus, it is surprising that over the course of the paper, none of the cell lines leveraged by the authors has been derived from a LUAD (MLE12 cells are mouse normal lung epithelial cells, H82 is a human SCLC cell line, and LLC1 is a mouse lung epidermoid (squamous) cell line, to my knowledge).

- Unfortunately, the functional data supporting the metastatic potential of RNF20 haploinsufficiency is pretty weak, and again not performed in the right cell lines, in my opinion. Additional evidence would be required to fully support a role for RNF20 haploinsufficiency in metastasis and patient survival impact, such as:

o In vivo metastasis experiment (intracardiac injections?)

o Assessment of RNF20 expression in LUAD primary tumors versus metastasis

o Using human LUAD cell lines

o Leveraging a GEMM LUAD model which whichever molecular driver, comparing wt versus Rnf20 \pm mice in terms of number of metastasis, latency for metastasis, etc.

- In figure 4, metabolic changes are mostly assessed (1) in vitro, using a mouse normal lung epithelial cell line instead of human lung cancer (LUAD) cell lines; or (2) in lung tissue of control versus Rnf20 \pm mice. Thus, it is hard to conclude whether these changes driven by RNF20 downregulation may occur in a tumoral context. Isogenic LUAD cell lines or GEMMs would be required to make that point with confidence.

- Correlations between RNF20 expression and glycolysis genes, even if significant, are pretty weak. Additional publicly available or in-house LUAD cohorts should be used to prove correlation of RNF20 expression with glycolysis genes or survival.

- It would be important that the authors perform few experiments with a pharmacological approach in vivo to see if there is any actual therapeutic potential of the findings. Otherwise, the clinical implications of the findings are weak or unclear.

Minor comments

- In Figure 5A, the control condition doesn't match for the legend and the figure

Reviewer #2

(Remarks to the Author)

Liu et al. demonstrated that depletion of a single copy of Rnf20 results in the formation of lung tumors in mice. The study revealed that this depletion leads to induced DNA damage, increased cell growth, epithelial-mesenchymal transition (EMT), and defects in metabolic gene transcription mediated by HIF1 α -induced Pol II pausing. These findings provide intriguing observations and compelling mechanistic insights. I strongly support the publication of this work, with the following minor issues addressed:

1. It would be valuable to analyze the consequences of Pol II pausing by performing combined analyses with Rnf20 \pm RNA-seq data. Is there down-regulation of gene expression resulting from the defects in Pol II pause release? Correlating changes in the pausing index with gene expression could shed light on this aspect. Additionally, using gene body Pol II ChIP-seq signals or changes in H2Bub ChIP-seq as a control might prove useful.
2. It is important to map the genomic locations of Rnf20 and HIF1 α . How many co-localizations can be identified? Do their specific bindings explain the defects in Pol II pause release?
3. To further strengthen the evidence for Pol II pausing, it would be beneficial to validate the total Pol II ChIP-seq results with qPCR. Moreover, performing pSer5 and pSer2 ChIP-seq experiments could provide additional mechanistic insights.

Reviewer #3

(Remarks to the Author)

The manuscript by Liu, Tang, Singh and et al describes some interesting observations resulting from Rnf20 haploinsufficiency. The spontaneous development of lung cancers is quite striking. The authors' work points to a dual mechanism for the development of the tumours, relating to defective DSB repair and HIF-1 mediated transcriptional reprogramming. Below are some considerations that I think would improve the manuscript and help contextualise their findings.

- 1) The authors show convincingly that Rnf20 \pm leads to increased foci of DNA damage and defective repair, alongside a HIF-1 transcriptional signature. While these are both seemingly dependent on the level of Rnf20, there are not necessarily as interconnected as suggested, particularly from the title and abstract. Better to state that in combination they may predispose to tumour development, as there is insufficient evidence to show the DNA damage is due to HIF-1. Likewise, a conditional Rnf \pm / HIF1 $-/-$ mouse model would be needed to test the contribution of targeting HIF-1 in vivo.
- 2) The reduced Pol II pausing with increased HIF-1 levels may just be a consequence of increased HIF transcription. I think this is to some extent what the authors are implying, and would fit with existing data (PMID 35031618, 23746844). To better describe this, they could ChIP for Pol II at HIF target genes in the MLE12 cells following treatment with PHD or VHL inhibitors? Presumably this may overcome the effect of Rnf20 haploinsufficiency?
- 3) The data suggests that H2Bub by Rnf20 is unlikely to be important for HIF-1 transcription. However, prior studies have implicated an H2Bub requirement for H3K4me3 and for HIF-1-mediated transcription (19410543, 34155378). This difference may relate to a distinction in HIF-1 complexes, where glycolytic genes tend to involve CDK mediator and Pol II pausing (PMID 35031618, 23746844), whereas other HIF-1 targets do not, which could explain their metabolic phenotype, and could be explored further. Examining angiogenesis may provide initial insights into this. To help determine the contribution of H2Bub, it would be helpful to see if they observe similar effects with Rnf40 or Ube2A on HIF-1 protein levels and EMT (e.g. in MLE12 cells).
- 4) The changes in Pol II would be strengthened by looking at Pol II Ser2 and Ser5 phosphorylation as markers of transcriptional initiation and elongation.

Minor points:

- Why are two quite distinct lung cancer types driven by Rnf20 loss? Some further discussion on this point may be helpful.
 - Rnf20 depletion by CRISPR and generation of clones. Please clarify the number of clones used, whether reconstitution corrected the phenotype, and it would be helpful to show PCR confirmation of the deletions and that they are haploinsufficient.
- It is unclear why hypoxia does not result in a more substantial increase in HIF-1 protein levels in the MLE12 cells (Fig 4b). PHD or VHL inhibition may help show that this pathway is fully functional in the parental cell type.

Reviewer #4

(Remarks to the Author)

Liu, Dobrova and colleagues here present their analysis of Rnf20 E3 ubiquitin ligase in lung cancer.

They show that Rnf20 heterozygote mice show markedly increased lung cancer incidence (adenocarcinoma and small cell). Rnf20 heterozygosity impaired DNA damage response signalling and resolution in CRISPR-targeted cell lines and in lung

samples from the mice. This heterozygosity also led to phenotypic changes consistent with epithelial-mesenchymal transition (EMT), along with altered expression of key EMT-related genes. Among the major pathways impacted by Rnf20 heterozygosity were those controlled by the HIF-1 alpha transcription factor. The authors demonstrate greatly increased Hif-1 alpha levels in Rnf20 heterozygous cells, along with increases in glycolytic enzymes and activities. They then show that major phenotypes of the Rnf20 heterozygote cells can be suppressed by siRNA knockdown of Hif1-alpha. Seeking a mechanism for the role of Hif1-alpha in determining the outcome of Rnf20 heterozygosity, the authors found altered transcriptional activity of RNA Pol II, dependent on Rnf20-Hif-1 alpha, along with alterations in histone H2B monoubiquitination that indicated another route by which Rnf20 heterozygosity impacted gene regulation. The study concludes with the demonstration of a negative correlation between Hif1-alpha target expression levels and RNF20 in lung cancer patients, linking the paper's main findings with clinical outcomes.

This is a credible study that should be of interest to a broad readership. It has potential clinical ramifications, as well as opening new avenues for further investigation of the biology of Rnf20. My comments pertain mainly to technical controls and clarifications.

Major comments

1. Supplementary Fig. 1 should include formal ('genomic') demonstration of the successful targeting of the Rnf20 locus. Similarly, data confirming the expected disruption of the Rnf20 locus by CRISPR should be presented.
2. Details of the generation of the LLC1 and H82 Rnf20/ RNF20 heterozygotes should be provided (methodological information and controls for the disruption). This may not be needed for H82 cells if the description for Fig. 2e is incorrect.
3. Rescue experiments should be presented for the MLE12 CRISPR cell line results. Performing these controls for the DNA damage response, proliferation and Hif-1 stabilisation experiments would exclude the possibility of an off-target impact or a clonal effect.
4. A rescue control experiment should be presented for the Hif-1 RNAi cell line data.
5. Is Hif-1 alpha monoubiquitinated? This should be tested as a potential direct link to Rnf20 that might indicate a mechanism for the high Hif-1 levels seen.
6. Given the complexity of Rnf20's roles in different tissues, more information on the current understanding of the roles of Rnf20 in cancer would be useful, either in the Introduction or in the Discussion. There are a number of recent papers that could be considered in the light of the results presented here, e.g. Duan et al. Nat. Commun. 2016 PMID: 27557628; Wegwitz et al. Cell Death Dis 2020 PMID: 33070155; Wang et al. Front Oncol. 2020 PMID: 33364200.

Minor comments

7. The legend for Supplementary Fig. 2e indicates that the H82 experiment uses siRNA to deplete RNF20, but the figure suggests that these are a heterozygote cell line. This should be clarified.
8. The expected Mendelian number of null Rnf20 mice from the heterozygote crosses performed should be provided, in support of the finding that no Rnf20(-/-) offspring were obtained.
9. All immunoblots should include size markers.
10. Statistical outcome for Supplementary Fig. 1d should be indicated.
11. The legend to Fig. 3h should specify that these are the top 10 GO terms (if this is the case), i.e. that these are not selected from the Metascape analysis.
12. It should be clarified what the 2 lanes shown for each genotype in Fig. 4b represent (replicates, separate subclones, etc.?).
13. It should be clarified what the gene expression changes are measured against in the heat map shown in Fig. 4d; what are the 3 lanes showing?
14. Technical/ methodological details should be provided of the LC-MS experiments shown in Fig. 4d.
15. The immunoblot shown in Fig. 5g should be of better quality.
16. 'RRPM' should be explained in the legend to Fig. 6.
17. The curves should be placed in front of the individual datapoints in the graphs shown in Fig. 7a.

Version 1:

Reviewer comments:

Reviewer #1

(Remarks to the Author)

The authors have satisfactorily addressed the criticisms raised.

Reviewer #2

(Remarks to the Author)

The authors have fully addressed my questions.

Reviewer #3

(Remarks to the Author)

The authors have thoroughly addressed all concerns raised and I would like to congratulate them on this interesting work.

Reviewer #4

(Remarks to the Author)

The revision of the study by Liu, Tang, Singh, Dobrevá and colleagues convincingly demonstrates a clear role for the E3 ligase, RNF20, in modulating HIF1-controlled gene regulation, leading to increased levels of lung cancer in Rnf20 heterozygote mice, as well as alterations in Rnf20 heterozygote cell lines that are consistent with the authors' proposed mechanisms.

The authors have performed a very thorough revision that has addressed in detail the comments made by me and the other referees. The data support the authors' conclusions and title. I am enthusiastic about this work and consider that it shows new insight into multiple pathways of tumour suppression by RNF20.

We would like to express our sincere gratitude to all the reviewers for their appreciation of our work and especially for their thoughtful and constructive comments, which helped us to improve the quality of our manuscript considerably and to clarify a number of important points. To address the reviewers' concerns we have performed a number of additional experiments, as detailed in the following point-by-point response.

Reviewers' comments are in italic:

Reviewer #1 (Remarks to the Author):

Liu, Tang, Singh et al. establish a lung cancer genetically-engineered mouse model (GEMM) with Rnf20 haploinsufficiency giving rise to tumors resembling lung adenocarcinoma and small cell carcinoma. Rnf20 inactivation leads to increased DNA damage, HIF1alpha upregulation and metabolic reprogramming, and to increased EMT and migratory phenotype in vitro, potentially as a consequence of HIF1alpha effect at pausing transcription of genes involved in metabolism and EMT, among others.

Overall, even if the manuscript is well-written and the experiments well designed, I am not sure that the right models were used to support conclusions, and the clinical implications are unclear. There is a number of concerns:

Response: We thank the reviewer for the constructive comments, which helped us to improve our manuscript considerably.

Major comments

- Figures 2A and 2B are not really supportive of authors conclusions, as in each of these, only one cell line versus another cell line is compared. Authors should provide data on additional datasets containing mRNA data of normal lung and LUAD. Also, if the authors want to make any claim regarding the potential of RNF20 downregulation as driver of LUADs, they should comment on the dispersion of RNF20 expression in LUAD, suggestive that that might be the case only in a subset of LUADs.

Response: We thank the reviewer for highlighting the importance of better addressing the clinical implications of our findings. To strengthen this aspect, we have added western blot analyzes of A549, A427 and H322 adenocarcinoma cells as well as H82 and H69 small-cell lung cancer (SCLC) cells compared to normal human bronchial epithelial cells. Additionally, we have included two new figures demonstrating the functional impact of RNF20 loss in A549 adenocarcinoma, H82 SCLC, and LCC1 poorly differentiated carcinoma cells.

Moreover, immunohistochemistry on lung tissue arrays revealed a progressive loss of RNF20 in adenocarcinoma patients and significantly reduced RNF20 protein levels in SCLC patients (Fig. 1d, 1e).

Fig. 1. d Immunohistochemistry of representative tissue samples from different types and grades of lung tumors from a tissue microarray stained with an anti-RNF20 antibody. Scale bars, 100 μ m. **e** Relative staining intensity (H-Score) for RNF20 in lung tumors of different types and grades. n=70 for AD (n=20 Grade 1, n=30 Grade 2, n=20 Grade 3); n=22 for SCLC and n=20 for normal lung (NL).

We also would like note that existing TCGA data are derived from tissue samples with varying cell compositions. Myeloid and mast cells, which highly express *RNF20*, may contribute to the observed variability in *RNF20* expression in LUAD (Reviewer Figure R1 below).

Reviewer Figure R1: Dot plot of *RNF20* expression level and frequency in adenocarcinoma patients (GSE131907).

- Regarding the clinical impact of the findings, by the data shown by the authors it seems that *RNF20* downregulation may only have a clinically relevant effect in LUAD (Figure 1G) but not in other lung tumor types, including SCLC (Figure S1F).

Response: We apologize for not sufficiently clarifying the presented data. Figure S1F (now Supplementary Fig. 1i and 1j in the revised manuscript) shows *RNF20* expression and overall survival in patients with lung squamous cell carcinoma (SCC), not in small cell lung cancer (SCLC) patients. We do observe significantly reduced RNF20 protein levels in SCLC patients, as shown above (Fig. 1h, 1i). Additionally, lungs from *RNF20*^{+/-} heterozygous mice develop lesions with morphological and immunohistochemical features resembling SCLC (Fig. 1a-c, Supplementary Fig. 1h).

Supplementary Fig. 1i Normalized expression of *RNF20* in normal lung and lung squamous cell carcinoma (SCC) tissues in TCGA datasets. FPKM stands for fragments per kilobase of exon per million mapped fragments. **j** Overall survival (Kaplan-Meier plot) of lung squamous cell carcinoma patients expressing high vs low levels of *RNF20*.

Thus, it is surprising that over the course of the paper, none of the cell lines leveraged by the authors has been derived from a LUAD (MLE12 cells are mouse normal lung epithelial cells, H82 is a human SCLC cell line, and LLC1 is a mouse lung epidermoid (squamous) cell line, to my knowledge).

Response: We selected MLE12 cells because they represent lung epithelial cells, allowing us to observe the effects of *Rnf20* heterozygous loss in a non-cancerous context. H82 cells, which exhibit reduced expression of *RNF20*, are derived from SCLC. As mentioned above, SCLC patients also show reduced *RNF20* levels compared to non-cancerous lung tissue. We used LLC1 cells, because they are typically considered poorly differentiated carcinoma cells without distinct characteristics of any specific subtype, and have been extensively used in studying lung cancer and metastasis. However, we agree with the reviewer that incorporating human adenocarcinoma cell lines would enhance the relevance of our findings. To address this, we have included analysis using A549 cells in both *in vitro* and *in vivo* assays (new Figures 6 and 7). These assays validate our findings and include assessments of tumor growth and metastasis *in vivo*. Consistent with findings in MLE12 cells, we observed that the ablation of a single *RNF20* allele in the human A549 adenocarcinoma cell line led to increased levels of HIF1 α and γ H2AX (Fig. 6a, 6b). Further, *RNF20* loss was sufficient to enhance clonogenic growth, migration, glucose uptake, glycolysis, and lactate secretion, effects that could all be suppressed by HIF1A silencing (Fig. 6c-h). Moreover, inhibition of glucose uptake significantly reduced the enhanced clonogenic growth and migration abilities of *RNF20*^{+/-} A549 cells (Fig. 6i, 6j). Consistent findings were observed with siRNA-mediated silencing of *RNF20* in the human small-cell lung cancer cell line H82 (Fig. 6k-m), as well as with the ablation of a single *Rnf20* allele in murine Lewis lung carcinoma (LLC1) cells (Supplementary Fig. 5a-k).

The concordant results in both mouse and human cell lines further underscore the significance of our conclusions.

Fig. 6. HIF1α activation upon RNF20 loss promotes cell growth and migration in human adenocarcinoma and small cell lung cancer cells

a Schematic representation of the strategy used to generate *RNF20*^{+/-} A549 cells and PCR genotyping results showing distinct bands for WT and *RNF20* KO alleles in control and *RNF20*^{+/-} A549 cells. **b** Western blot analysis for RNF20, HIF1α, γH2AX and α-Tubulin of total cell lysates of control (*RNF20*^{+/+}) and *RNF20*^{+/-} A549 cells. **c-e** Clonogenic (**c**) and Boyden chamber migration assay (**d**) with control and *RNF20*^{+/-} A549 cells infected either with control shRNA or shRNA against *HIF1a* and quantification (**e**). **f** 2-DG uptake (top) and concentration of lactate in the supernatant (bottom) of control and *RNF20*^{+/-} A549 cells stably expressing control shRNA or shRNA against *HIF1A*. **g, h** qPCR analysis of the glucose transporter *SLC2A1* (*GLUT1*) and *LDHA* (**g**) and ECAR (**h**) in control and *RNF20*^{+/-} A549 cells, stably expressing either control shRNA or shRNA against *HIF1A*. **i** Clonogenic assay with control and

RNF20^{+/-} A549 cells treated either with DMSO or with WZB117 (left) and quantification (right). **j** Boyden chamber migration assay with control and *RNF20*^{+/-} A549 cells treated either with DMSO or with WZB117 (left) and quantification of the number of colonies (right). Scale bars, 150µm. **k, l** Quantification of the number of colonies in colony formation assay (**k**) and migrated cells per field in a Boyden chamber migration assay (**l**) with H82 cells transfected with control siRNA or siRNA against *RNF20* treated either with DMSO or with WZB117. **m** Clonogenic assay with H82 cells transfected with control siRNA or siRNA against *RNF20* treated with DMSO or with WZB117. Statistical analysis in (**f, k, l**) was performed using a two-tailed Student's t-test. Multiple comparisons in (**e, g, h, i, j, m**) were performed using ANOVA. Data are shown as means ± SEM. *p<0.05, **p<0.01, ***p<0.001.

- Unfortunately, the functional data supporting the metastatic potential of *RNF20* haploinsufficiency is pretty weak, and again not performed in the right cell lines, in my opinion. Additional evidence would be required to fully support a role for *RNF20* haploinsufficiency in metastasis and patient survival impact, such as:

Response: We agree with the reviewer and have performed the requested experiments as follows.

o In vivo metastasis experiment (intracardiac injections?)

To further investigate the role of *RNF20* loss in tumor growth and metastatic dissemination *in vivo*, we injected control and *RNF20*^{+/-} A549 cells subcutaneously and intravenously into nude mice. Tumor volume in mice transplanted with *RNF20*^{+/-} A549 cells was already significantly larger 9 days post-injection and continued to increase thereafter (Fig. 7a, 7b). Similarly, intravenous injection of *RNF20*^{+/-} A549 cells resulted in a significantly higher number of tumor nodules and an expanded metastatic area in the lungs compared to mice injected with control A549 cells (Fig. 7c, 7d).

Consistent with these findings, C57BL/6 mice injected intravenously with *Rnf20*^{+/-} LLC1 cells also exhibited a significantly increased number of pulmonary tumor nodules and a larger metastatic area compared to controls (Fig. 7e, 7f).

Importantly, in line with our cell culture-based assays, *HIF1A* silencing or inhibition of glucose uptake with the GLUT1 inhibitor WZB117 was sufficient to suppress the metabolic changes (Fig. 7g) and reduce both the number of tumor nodules and the metastatic area in mice injected with *RNF20*^{+/-} A549 cells (Fig. 7h-j). Notably, tumors from mice injected with *RNF20*^{+/-} A549 cells exhibited significantly elevated γH2AX levels compared to those in control mice, an effect that was reversed by *HIF1A* silencing or glucose uptake inhibition (Fig. 7h, 7k). Taken together, these data suggest a critical role of HIF1α-mediated metabolic rewiring in the increased tumor growth and metastasis upon *RNF20* LOF *in vivo*.

Fig. 7. RNF20 loss promotes tumor growth and metastasis through HIF1 α activation and metabolic rewiring. **a** Control and *RNF20*^{+/-} A549 cells were injected subcutaneously into the flanks of BALB/C Nude mice (n=6). Mice were sacrificed after 27 days and the subcutaneous tumors were removed. **b** Quantification of the macroscopic tumor volume at different days after injection. **c** Control and *RNF20*^{+/-} A549 cells were injected intravenously into the tail vein of BALB/C Nude mice (n=6). Macroscopic appearance (upper panels) and HE staining (lower panels) of representative lungs. Scale bars, 2 mm. **d** Quantification of the metastatic nodules (upper panels) and the metastasis area (lower panels), in lung sections of mice injected with control and *RNF20*^{+/-} A549 cells. **e** Control and *Rnf20*^{+/-} LLC1 cells were injected intravenously into the tail vein of C57BL/6 mice (n=6). Macroscopic appearance (upper panels) and HE staining (lower panels) of representative lungs. Scale bars, 2 mm. **f** Quantification of the metastatic nodules (upper panels) and the metastasis area (lower panels), in lung sections of mice injected with control and *Rnf20*^{+/-} LLC1 cells. **g** Concentration of glucose-6-phosphate (G6P, top

panel) and lactate in lung homogenates (bottom panel) of BALB/C Nude mice injected intravenously into the tail vein with control and *RNF20*^{+/-} A549 cells, as well as control and *RNF20*^{+/-} A549 cells after shRNA mediated *HIF1A* silencing or after treatment with the GLUT1 inhibitor WZB117 (n=5). **h** Macroscopic appearance (upper panels) and γ H2AX combined with HE staining (lower panels) of representative lungs. Scale bars, 2 mm. **i-k** Quantification of the metastatic nodules (**i**), metastasis area (**j**) in lung sections and the relative staining intensity (H-Score) for γ H2AX (**k**) in lung tumors of BALB/C Nude mice injected with control and *RNF20*^{+/-} A549 cells, as well as control and *RNF20*^{+/-} A549 cells after shRNA mediated *HIF1A* silencing or after treatment with the GLUT1 inhibitor WZB117 (n=5). Statistical analysis in (**b**, **d**, **f**) was performed using a two-tailed Student's t-test. Multiple comparisons in (**g**, **i**, **j**, **k**) were performed using ANOVA. Data are shown as means \pm SEM. * $p < 0.05$, ** $p < 0.01$, *** $p < 0.001$.

o Assessment of RNF20 expression in LUAD primary tumors versus metastasis

To further investigate the relationship between RNF20 levels in primary tumors and metastases, we analyzed lung sections from two experimental models². In the first model, mice received an intratracheal injection of LLC1 cells. In the second, a spontaneous metastasis model with tumor resection (surgical resection model), LLC1 cells were injected subcutaneously to establish a primary tumor, which was surgically removed after 10 days to allow for metastasis relapse. Importantly, RNF20 levels were significantly higher in primary tumors, such as those from the intratracheal injection model, compared to tumors in the surgical resection model (Fig. 1j, 1k).

Fig. 1j Immunohistochemistry of lungs from intratracheal (i.t) injection model and surgical resection model stained with an anti-RNF20 antibody. Scale bar, 400 μ m. **Fig. 1k** Relative staining intensity (H-Score) of RNF20 in i.t and tumor relapse groups (n=5).

o Using human LUAD cell lines

As mentioned in the response above, we have included two new figures that analyze the role of *RNF20* loss in human A549 adenocarcinoma cells (Figures 6 and 7).

o Leveraging a GEMM LUAD model which whichever molecular driver, comparing wt versus Rnf20^{+/-} mice in terms of number of metastasis, latency for metastasis, etc.

Since the deletion of *Rnf20* leads primarily to the formation of tumors with characteristics of SCLC (Supplementary Fig. 1h), crossing the *Rnf20* knockout allele into a GEMM model of

LUAD may lead to results that are difficult to interpret. Moreover, many of the commonly used LUAD GEMMs form no or very few distant metastases. Therefore, in addressing the role of RNF20 in metastasis, we decided to focus on the other approaches suggested by the reviewer, as detailed in our responses to the preceding sub-points of this comment. Nevertheless, it is interesting to point out that histological examination also detected lesions in the livers of *Rnf20*^{+/-} heterozygous mice, which were positive for the calcitonin gene-related peptide (CGRP) and ASCL1 (Achaete-Scute Complex Homolog 1), markers for small cell lung cancer of neuroendocrine origin (Reviewer Figure R2). However, since our model involves a germline deletion of *Rnf20*, we are not able to unambiguously determine whether these neoplastic lesions represent metastases originating from the lungs or locally arising tumors.

Reviewer Figure R2: Neoplastic lesions in the liver of *Rnf20*^{+/-} mice express markers of SCLC. Immunohistochemistry of livers from *Rnf20*^{+/-} mice stained with an CGRP and ASCL1 antibody.

- In figure 4, metabolic changes are mostly assessed (1) *in vitro*, using a mouse normal lung epithelial cell line instead of human lung cancer (LUAD) cell lines; or (2) in lung tissue of control versus *Rnf20*^{+/-} mice. Thus, it is hard to conclude whether these changes driven by RNF20 downregulation may occur in a tumoral context. Isogenic LUAD cell lines or GEMMs would be required to make that point with confidence.

Response: We agree with the reviewer and have performed the requested experiments. Consistent with our results in MLE12 cells, we observed that the ablation of a single *RNF20* allele in the human A549 adenocarcinoma cell line led to enhanced glucose uptake, glycolysis, and lactate secretion, effects that could all be suppressed by HIF1A silencing (Fig. 6f-h).

Fig. 6f 2-DG uptake (top) and concentration of lactate in the supernatant (bottom) of control and *RNF20*^{+/-} A549 cells stably expressing control shRNA or shRNA against *HIF1A*. **6g, 6h** qPCR analysis

of the glucose transporter *SLC2A1* (*GLUT1*) and *LDHA* (6g) and ECAR (6h) in control and *RNF20*^{+/-} A549 cells, stably expressing either control shRNA or shRNA against *HIF1A*.

In line with our cell culture-based assays, *HIF1A* silencing or inhibition of glucose uptake with the GLUT1 inhibitor WZB117 was sufficient to suppress the metabolic changes (Fig. 7g) in mice injected with *RNF20*^{+/-} A549 cells compared to mice injected with control A549 cells (Fig. 7g).

Fig. 7g Concentration of glucose-6-phosphate (G6P, left panel) and lactate in lung homogenates (right panel) in BALB/C Nude mice injected intravenously into the tail vein with control and *RNF20*^{+/-} A549 cells, as well as control and *RNF20*^{+/-} A549 cells after shRNA mediated *HIF1A* silencing or after treatment with the GLUT1 inhibitor WZB117 (n=5).

- Correlations between *RNF20* expression and glycolysis genes, even if significant, are pretty weak. Additional publicly available or in-house LUAD cohorts should be used to prove correlation of *RNF20* expression with glycolysis genes or survival.

Response: To further explore the correlation between *RNF20* levels with the expression of HIF1 α and HIF1 α -dependent metabolic enzymes in lung cancer patients we stained lung cancer tissue microarray with HIF1 α , ENO1 and LDHA antibodies. We observed significantly increased protein levels of HIF1 α , ENO1, and LDHA in patients with SCLC and high-grade AD, which were inversely correlated with *RNF20* protein levels (Fig. 10a, 10b).

Fig. 10. Decreased RNF20 levels correlate with increased levels of HIF1 α and glycolytic enzymes in AD and SCLC patients. **a** Immunohistochemistry of representative tissue samples from different types and grades of lung tumors from a tissue microarray stained with HIF1 α , ENO1 and LDHA antibody. Scale bars, 100 μ m. **b** Pearson correlation (r) of the relative staining intensity (H-Score) of RNF20 and HIF1 α (top panels), RNF20 and ENO1 (middle panels), and RNF20 and LDHA (bottom panels) in AD and SCLC patients. $n=70$ for AD; $n=22$ for SCLC.

- It would be important that the authors perform few experiments with a pharmacological approach *in vivo* to see if there is any actual therapeutic potential of the findings. Otherwise, the clinical implications of the findings are weak or unclear.

Response: We agree with the reviewer and have performed the requested experiments. Consistent with our cell culture-based assays, *HIF1A* silencing or inhibition of glucose uptake with the GLUT1 inhibitor WZB117 was sufficient to suppress the metabolic changes (Fig. 7g) and reduce both the number of tumor nodules and the metastatic area in mice injected with *RNF20*^{+/-} A549 cells (Fig. 7h-j). Notably, tumors from mice injected with *RNF20*^{+/-} A549 cells exhibited significantly elevated γ H2AX levels compared to those in control mice, an effect that was reversed by *HIF1A* silencing or glucose uptake inhibition (Fig. 7h, 7k).

Figure 7

Fig. 7g Concentration of glucose-6-phosphate (G6P, top panel) and lactate in lung homogenates (bottom panel) of BALB/C Nude mice injected intravenously into the tail vein with control and *RNF20*^{+/-} A549 cells, as well as control and *RNF20*^{+/-} A549 cells after shRNA mediated *HIF1A* silencing or after treatment with the GLUT1 inhibitor WZB117 ($n=5$). **7h** Macroscopic appearance (upper panels) and γ H2AX combined with HE staining (lower panels) of representative lungs. Scale bars, 2 mm. **7i-k** Quantification of the metastatic nodules (**i**), metastasis area (**j**) in lung sections and relative staining intensity (H-Score) for γ H2AX (**k**) in lung tumors of BALB/C Nude mice injected with control and *RNF20*^{+/-} A549 cells, as well as control and *RNF20*^{+/-} A549 cells after shRNA mediated *HIF1A* silencing or after treatment with the GLUT1 inhibitor WZB117 ($n=5$).

Minor comments

- *In Figure 5A, the control condition doesn't match for the legend and the figure*

Response: We apologize for the inconsistencies and have corrected the figure.

Reviewer #2 (Remarks to the Author):

Liu et al. demonstrated that depletion of a single copy of *Rnf20* results in the formation of lung tumors in mice. The study revealed that this depletion leads to induced DNA damage, increased cell growth, epithelial-mesenchymal transition (EMT), and defects in metabolic gene transcription mediated by HIF1a-induced Pol II pausing. These findings provide intriguing observations and compelling mechanistic insights. I strongly support the publication of this work, with the following minor issues addressed:

Response: We thank the reviewer for the commendation of our work and for the constructive comments.

1. It would be valuable to analyze the consequences of Pol II pausing by performing combined analyses with *Rnf20*^{+/-} RNA-seq data. Is there down-regulation of gene expression resulting from the defects in Pol II pause release? Correlating changes in the pausing index with gene expression could shed light on this aspect. Additionally, using gene body Pol II ChIP-seq signals or changes in H2Bub ChIP-seq as a control might prove useful.

Response: We thank the reviewer for this comment. We found a significant decrease in the pausing index at genes upregulated in *Rnf20*^{+/-} versus control MLE12 cells, while we did not detect major changes at genes that were not changed or downregulated (Fig. 9c).

Fig. 9c

Fig. 9c Log2 of the PI at genes upregulated, unchanged or downregulated upon *Rnf20* LOF in control and *Rnf20*^{+/-} MLE12 cells.

We have also performed a correlation analysis, showing a significant negative correlation between the RNA Pol II pausing index and gene expression for upregulated genes, but not for downregulated genes (Supplementary Fig. 6b). Furthermore, we did not detect significant correlation between PI and H2Bub1 changes, indicating that Pol II pausing is independent of H2Bub1 levels (Supplementary Fig. 6d).

Supplementary Fig. 6b Pearson correlation (r) for the Log₂ FC of PI and Log₂ FC of upregulated (left) or downregulated (right) genes in *Rnf20*^{+/-} versus control MLE12 cells. **6d** Pearson correlation (r) for the Log₂ FC of PI and Log₂ FC of genes showing increased (left) or decreased (right) H2Bub1 levels in *Rnf20*^{+/-} versus control MLE12 cells.

2. It is important to map the genomic locations of *Rnf20* and *HIF1a*. How many co-localizations can be identified? Do their specific bindings explain the defects in Pol II pause release?

Response: We thank the reviewer for this comment, which has helped us to clarify an important point. Since HIF1 α has been shown to stimulate Pol II pause release, we intersected HIF1 α -bound genes in lung adenocarcinoma cells¹ with genes exhibiting decreased PI upon RNF20 loss. GO analysis revealed that the overlapping genes are associated with the mRNA metabolism, HIF-1 signaling pathway, glycolytic processes and mechanisms associated with pluripotency (Fig. 9h). Consistent with a direct role of HIF1 α in regulating Pol II pause release, the reduced Pol II pausing index in *Rnf20*^{+/-} MLE12 cells at HIF1 α bound genes was increased to control levels upon HIF1 α depletion (Fig. 9i, 9j).

Fig. 9h Venn diagram showing the overlap of genes with decreased PI in *Rnf20*^{+/-} compared to control MLE12 cells and HIF1 α -bound genes in A549 lung AD cells, determined by ChIP-Seq¹ (left) and GO analysis of the overlapping genes (right). **i** Genome tracks of merged Pol II ChIP-Seq reads of control, *Rnf20*^{+/-} and *Rnf20*^{+/-} MLE12 cells stably expressing shRNA against *Hif1a*. **j** PI of control, *Rnf20*^{+/-} and *Rnf20*^{+/-} MLE12 cells stably expressing shRNA against *Hif1a* at HIF1 α -bound genes in lung A549 AD cells.

Taken together these data suggest that HIF1 α activation upon *Rnf20* haploinsufficiency induces RNA polymerase II promoter-proximal pause release at HIF1 α -target genes.

3. To further strengthen the evidence for Pol II pausing, it would be beneficial to validate the total Pol II ChIP-seq results with qPCR. Moreover, performing pSer5 and pSer2 ChIP-seq experiments could provide additional mechanistic insights.

Response: We thank the reviewer for this comment. Pol II ChIP-qPCR on glycolytic targets in control and *Rnf20*^{+/-} MLE12 cells as well as WT and *Rnf20*^{+/-} lung samples, confirmed the decreased Pol II pausing upon *Rnf20* loss, both *in vitro* and *in vivo* (Fig. 9g, Supplementary Fig. 6c).

Fig. 9g Ratio of the Pol II enrichment at the TSS and TTS of *Slc2a1*, *Eno1* and *Pdk1* in control and *Rnf20*^{+/-} lungs determined by Pol II ChIP-qPCR. TSS, Transcription Start Site; TTS, Transcription Termination Site. **Supplementary Fig. 6c** Ratio of the Pol II enrichment at the TSS and TTS of *Slc2a1*, *Eno1* and *Pdk1* in control and *Rnf20*^{+/-} MLE12 cells determined by Pol II ChIP-qPCR.

To further validate these findings and study the role of HIF1 α activation upon RNF20 loss, we performed ChIP-qPCR for initiating/paused and elongating Pol II enriched in phosphorylated serine 5 (pSer5) and serine 2 (pSer2), respectively, on glycolytic targets in control, *Rnf20*^{+/-}, and *Rnf20*^{+/-} HIF1 α knockdown MLE12 cells. We observed increased enrichment of elongating Pol II-pSer2 across the gene bodies of *Slc2a1* (*Glut1*), *Eno1*, and *Ldha* upon RNF20 loss, which was attenuated upon HIF1 α depletion (Fig. 9k, left panel). Concurrently, initiating/paused Pol II-pSer5 was reduced at the transcription start sites of these genes in *Rnf20*^{+/-} MLE12 cells, but was restored to control levels following HIF1 α depletion (Fig. 9k, right panel).

Fig. 9k Chip-qPCR for RNA Pol II CTD-pSer2 (left) and CTD-pSer5 (right) at *Slc2a1*, *Eno1* and *Pdk1* gene body in control, *Rnf20*^{+/-} MLE12 cells, or *Rnf20*^{+/-} MLE12 cells after shRNA mediated *Hif1a* silencing. Venn diagram showing an overlap of genes with decreased PI in *Rnf20*^{+/-} compared to control MLE12 cells and HIF1 α -bound genes in A549 lung AD cells, determined by CHIP-Seq¹ (left) and GO analysis of genes with decreased PI in *Rnf20*^{+/-} compared to control MLE12 cells and genes bound by HIF1 α (right).

Reviewer #3 (Remarks to the Author):

The manuscript by Liu, Tang, Singh and et al describes some interesting observations resulting from *Rnf20* haploinsufficiency. The spontaneous development of lung cancers is quite striking. The authors' work points to a dual mechanism for the development of the tumours, relating to defective DSB repair and HIF-1 mediated transcriptional reprogramming. Below are some considerations that I think would improve the manuscript and help contextualise their findings.

Response: We thank the reviewer for the constructive comments, which helped us to improve our manuscript considerably.

1) The authors show convincingly that *Rnf20* +/- leads to increased foci of DNA damage and defective repair, alongside a HIF-1 transcriptional signature. While these are both seemingly dependent on the level of *Rnf20*, there are not necessarily as interconnected as suggested, particularly from the title and abstract. Better to state that in combination they may predispose to tumour development, as there is insufficient evidence to show the DNA damage is due to HIF-1. Likewise, a conditional *Rnf* +/- HIF1 -/- mouse model would be needed to test the contribution of targeting HIF-1 in vivo.

Response: We thank the reviewer for this insightful comment, which has allowed us to further refine and strengthen the conclusion on this point. We observed increased γ H2AX under hypoxia, suggesting that HIF1 α activation may contribute to the increased DNA damage upon RNF20 loss. Indeed, *Hif1a* silencing was sufficient to reduce HIF1 α and γ H2AX levels in *Rnf20*+/- MLE12 cells to control levels (Fig. 5g). Notably, tumors from mice injected with RNF20+/- A549 lung adenocarcinoma cells exhibited significantly elevated γ H2AX levels compared to those developed in mice injected with control A549 cells, an effect that was reversed by HIF1A silencing or glucose uptake inhibition (Fig. 7k). Taken together, these data suggest that HIF1 α -mediated metabolic rewiring plays a critical role in the increased DNA damage observed upon RNF20 loss of function.

Fig. 5g Western blot analysis for HIF1 α and γ H2AX in total cell lysates of MLE12 cells under normoxic or hypoxic conditions (1%O₂) (top) and control, *Rnf20*+/- or *Hif1a* knockdown *Rnf20*+/- MLE12 cells in normoxic conditions (bottom). **7h** Macroscopic appearance (upper panels) and γ H2AX combined with HE staining (lower panels) of representative lungs of BALB/C Nude mice injected intravenously into the

tail vein with control and *RNF20*^{+/-} A549 cells, as well as control and *RNF20*^{+/-} A549 cells after shRNA mediated *HIF1A* silencing or after treatment with the GLUT1 inhibitor WZB117 (n=5). Scale bars, 2 mm. **7k** Relative staining intensity (H-Score) for γ H2AX in lung tumors of BALB/C Nude mice injected control and *RNF20*^{+/-} A549 cells, as well as control and *RNF20*^{+/-} A549 cells after shRNA mediated *HIF1A* silencing or after treatment with the GLUT1 inhibitor WZB117 (n=5).

2) *The reduced Pol II pausing with increased HIF-1 levels may just be a consequence of increased HIF transcription. I think this is to some extent what the authors are implying, and would fit with existing data (PMID 35031618, 23746844). To better describe this, they could ChIP for Pol II at HIF target genes in the MLE12 cells following treatment with PHD or VHL inhibitors? Presumably this may overcome the effect of Rnf20 haploinsufficiency?*

Response: We apologize for not articulating our findings more clearly. Indeed, our data suggest that the reduced Pol II pausing on HIF1 α target genes is a consequence of increased HIF1 α activity, as silencing of HIF1 α restored the pausing index to control levels. We have now included new data and revised the text, as follows: Since HIF1 α has been shown to stimulate Pol II pause release, we intersected HIF1 α -bound genes in lung adenocarcinoma cells¹ with genes exhibiting decreased PI upon RNF20 loss. GO analysis revealed that the overlapping genes are associated with the mRNA metabolism, HIF-1 signaling pathway, glycolytic processes and mechanisms associated with pluripotency (Fig. 9h). Consistent with a direct role of HIF1 α in regulating Pol II pause release, the reduced Pol II pausing index in *Rnf20*^{+/-} MLE12 cells at HIF1 α bound genes was increased to control levels upon HIF1 α depletion (Fig. 9i, 9j).

Fig. 9h Venn diagram showing the overlap of genes with decreased PI in *Rnf20*^{+/-} compared to control MLE12 cells and HIF1 α -bound genes in A549 lung AD cells, determined by ChIP-Seq¹ (left) and GO analysis of the overlapping genes (right). **i** Genomic tracks of merged Pol II ChIP-Seq reads of control, *Rnf20*^{+/-} and *Rnf20*^{+/-} MLE12 cells stably expressing shRNA against *Hif1a*. **j** PI of control, *Rnf20*^{+/-} and *Rnf20*^{+/-} MLE12 cells stably expressing shRNA against *Hif1a* at HIF1 α -bound genes in lung A549 AD cells.

Taken together these data suggest that HIF1 α activation upon *Rnf20* haploinsufficiency induces RNA polymerase II promoter-proximal pause release at HIF1 α -target genes.

3) The data suggests that H2Bub by Rnf20 is unlikely to be important for HIF-1 transcription. However, prior studies have implicated an H2Bub requirement for H3K4me3 and for HIF-1-mediated transcription (19410543, 34155378). This difference may relate to a distinction in HIF-1 complexes, where glycolytic genes tend to involve CDK mediator and Pol II pausing (PMID 35031618, 23746844), whereas other HIF-1 targets do not, which could explain their metabolic phenotype, and could be explored further. Examining angiogenesis may provide initial insights into this.

Response: We thank the reviewer for this comment. As the reviewer mentioned, RNF20 functions as an E3 ubiquitin ligase for H2Bub1, a modification that promotes SET1-dependent di- and trimethylation of H3K4³. SET1B is also required for the activation of HIF-inducible genes⁴. To further explore the role of HIF1 α in transcriptional regulation in lung epithelial cells, we analyzed the relationship between H2Bub1 changes and PI in *Rnf20*^{+/-} versus control MLE12 cells and performed ChIP-Seq for H3K4me3 in *Rnf20*^{+/-} and *Rnf20*^{+/-} HIF1 α knockdown MLE12 cells. The results showed no significant correlation between PI and H2Bub1 changes, indicating that Pol II pausing is independent of H2Bub1 levels (Supplementary Fig. 6d). Furthermore, we did not observe changes in H3K4me3 levels at HIF1 α target genes or genes upregulated in *Rnf20*^{+/-} cells upon HIF1 α silencing, but we identified a decrease in H3K4me3 at genes downregulated by RNF20 LOF (Supplementary Fig. 6e). These findings suggest that the HIF1 α -dependent reduction of H3K4me3 may contribute to the transcriptional downregulation of certain RNF20 target genes. However, this mechanism does not appear to play a role in the regulation of RNF20-mediated transcriptional pausing at HIF1 α target genes in lung epithelial cells.

To help determine the contribution of H2Bub, it would be helpful to see if they observe similar effects with *Rnf40* or *Ube2A* on HIF-1 protein levels and EMT (e.g. in MLE12 cells).

Response: We thank the reviewer for this comment, which has helped us to clarify a number of important points. We now present several lines of evidence suggesting that, at least in lung cancer, RNF20 has a distinct role compared to RNF40. As noted by the reviewer, RNF20 or RNF40 form a heterodimer that acts as the major E3 ligase responsible for histone H2Bub1 in mammalian cells⁵. Interestingly, in contrast to *RNF20* mRNA levels (Fig. 1f, 1g), *RNF40* mRNA levels were increased in AD patients (Fig. 1h) and higher *RNF40* levels significantly correlated with poor survival in the KMplotter lung adenocarcinoma dataset⁶ (Fig. 1i), suggesting distinct function of these two proteins in lung cancer. *Rnf40* silencing, in contrast to RNF20 LOF, did not affect HIF1 α levels (Fig. 5m), cell migration or clonogenic growth (Fig. 5n, 5o, Supplementary Fig. 4k, 4l), but significantly reduced the expression of some HIF1 α targets, such as *Fn1*, *Snai2*, *Vegfa* and *Eno1* (Fig. 5p). These findings indicate that RNF20 operates independently of RNF40 in suppressing cell growth, migration, and the regulation of HIF1 α -responsive genes, at least in lung epithelial and lung cancer cells.

Fig. 1f, h Normalized expression of *RNF20* (f) or *RNF40* (h) in normal lung and AD tissues in TCGA datasets. FPKM stands for fragments per kilobase of exon per million mapped fragments. **g, i** Overall

survival (Kaplan-Meier plot) of lung AD patients⁶ expressing high vs low levels of *RNF20* (probe 222683_at) (g) or *RNF40* (probe 206845_s_at) (i). **Fig. 5 m** Western blot analysis of RNF20, RNF40 and H2Bub1 in normoxic conditions, as well as HIF1 α under normoxic and hypoxic conditions, in total cell lysates from control or *Rnf40* knockdown MLE12 cells. **n** Quantification of the number of migrated cells per field in a Boyden chamber migration assay with control and *Rnf40* knockdown MLE12 cells. **o** Clonogenic assay with control and *Rnf40* knockdown MLE12 cells. **p** qPCR analysis of genes involved in EMT (left) or glycolysis (right) in control and *Rnf40* knockdown MLE12 cells. **Supplementary Fig. 4k** Representative images of Boyden chamber-based migration assay control or *Rnf40*-shRNA-mediated knockdown MLE12 cells. Scale bars, 100 μ m. **l** Representative images of clonogenic assay performed with control or *Rnf40*-shRNA-mediated knockdown MLE12 cells.

4) The changes in Pol II would be strengthened by looking at Pol II Ser2 and Ser5 phosphorylation as markers of transcriptional initiation and elongation.

Response: To further investigate the role of HIF1 α activation upon RNF20 loss and its impact on Pol II pausing and pause release, we performed ChIP-qPCR for initiating/paused and elongating Pol II enriched in phosphorylated serine 5 (pSer5) and serine 2 (pSer2), respectively, on glycolytic targets in control, *Rnf20*^{+/-}, and *Rnf20*^{+/-} HIF1 α knockdown MLE12 cells. We observed increased enrichment of elongating Pol II-pSer2 across the gene bodies of *Slc2a1* (*Glut1*), *Eno1*, and *Ldha* upon RNF20 loss, which was attenuated upon HIF1 α depletion (Fig. 9k). Concurrently, initiating/paused Pol II-pSer5 was reduced at the transcription start sites of these genes in *Rnf20*^{+/-} MLE12 cells, but was restored to control levels following HIF1 α depletion (Fig. 9k).

Fig. 9k Chip-qPCR for RNA Pol II CTD-pSer2 (left) and CTD-pSer5 (right) at *Slc2a1*, *Eno1* and *Pdk1* gene body in control, *Rnf20*^{+/-} MLE12 cells, or *Rnf20*^{+/-} MLE12 cells after shRNA mediated *Hif1a* silencing. Venn diagram showing a overlap of genes with decreased PI in *Rnf20*^{+/-} compared to control MLE12 cells and HIF1 α -bound genes in A549 lung AD cells, determined by ChIP-Seq¹ (left) and GO analysis of genes with decreased PI in *Rnf20*^{+/-} compared to control MLE12 cells and genes bound by HIF1 α (right).

Minor points:

-Why are two quite distinct lung cancer types driven by *Rnf20* loss? Some further discussion on this point may be helpful.

Response: We thank the reviewer for this comment. We have expanded and restructured the discussion on why RNF20 loss might specifically drive lung adenocarcinoma and small cell lung carcinoma (page 17, line 6 – page 19, line 6). There are two primary histopathological groups of lung tumors: SCLC and non-small-cell lung cancer (NSCLC). The latter accounts for approximately 80% of all cases, with adenocarcinoma being the most common type⁷. These different lung cancer types have distinct origins, clinical and pathological features⁸. SCLC is an aggressive type of lung cancer with neuroendocrine origin that typically grows and spreads rapidly. Alveolar type II (AT2) cells, which play a central role in maintaining the integrity and function of the alveoli, are the primary cells of origin of lung AD. While adenocarcinoma is the most common subtype, the marked increase in SCLC-like lesions could be attributed to the concomitant decrease in p53 and RB, along with elevated HIF1 α activity and decreased Notch signaling. Somatic inactivation of *Rb1* and *p53* using intratracheal injection or intubation of mice with adenovirus-Cre leads to the development of SCLC-like tumors⁹, whereas mice carrying germline deletion of one *Rb1* allele and both p53 alleles (*Rb1*^{+/-}*p53*^{-/-}) develop a variety of primary tumors of neuroendocrine origin as well as focal bronchial neuroendocrine cell hyperplasia¹⁰, showing the critical importance of these factors in preventing transformation and growth of neuroendocrine cells. Intriguingly, *RB1* knockdown in hESC-derived pulmonary neuroendocrine cells (PNEC) produces CGRP-expressing cells with gene expression characteristics of SCLC¹¹. Bi-allelic inactivation of *TP53* and *RB1* have been reported in nearly all SCLC¹², further supporting the notion that LOF of p53 and RB in *Rnf20* haploinsufficient mice could be responsible for the increased incidence of SCLC-like lesions in these mice. p53 is a known primary downstream target of RNF20¹³, whereas the effect of RNF20 on RB has so far not been examined. While we found a major decrease in H2Bub1 at genes downregulated in *Rnf20*^{+/-} cells, including *p53*, we did not observe profound changes in H2Bub1 at the *Rb1* locus (Fig. 6k), suggesting that the decrease of *Rb1* mRNA levels in *Rnf20* haploinsufficient cells and animals is independent of transcriptional regulation via the Rnf20-H2bub1 axis. Furthermore, we detected significant RNF20-H2Bub1-dependent decrease in the expression of *Rbx1*, a key component of the VHL-containing ubiquitin ligase complex that initiates the degradation of HIF1 α ¹⁴. As a consequence, HIF1 α protein, but not mRNA, levels were elevated in RNF20 LOF cells even under normoxic conditions, thereby disrupting oxygen sensing and creating a state of pseudohypoxia. In this context, it is noteworthy that PNECs, which are considered cells of origin for SCLC, play a crucial role as oxygen sensors. Abnormal oxygen sensing, caused by the loss of prolyl hydroxylase domain proteins (PHDs) function, regulating the stability of HIF-1 α , leads to PNEC hyperplasia¹⁵. Moreover, *Hes1* showed significant downregulation, which was accompanied by a decrease in H2bub1 levels. Similarly, RNF20 is required for Notch-dependent gene expression in endothelial cells¹⁶, and its *Drosophila* homolog, dBre1, has been shown to regulate the expression of Notch target genes

by coupling H2Bub1 to H3K4me3¹⁷. Notch signaling suppresses SCLC, as genetic and pharmacological inhibition of Notch activity were associated with an increased tumor number¹², suggesting that decreased Notch signaling may play a role in SCLC development upon *Rnf20* loss.

- *Rnf20* depletion by CRISPR and generation of clones. Please clarify the number of clones used, whether reconstitution corrected the phenotype, and it would be helpful to show PCR confirmation of the deletions and that they are haploinsufficient.

Response: We have revised the Materials and Methods section to provide a clearer explanation of the generation of clones in the various mouse and human lung epithelial and lung cancer cell lines. The sample sizes (n) presented in the plots represent individual clones. We have now provided formal evidence of successful targeting of the *Rnf20* locus in Supplementary Fig. 1c. Similarly, PCR analysis confirming the expected deletion of the *RNF20/Rnf20* locus by CRISPR is presented in Fig. 6a (A549), Supplementary Fig. 2a (MLE12 cells), and Supplementary Fig. 5a (LLC1 cells).

Supplementary Fig. 1a Schematic diagram of the *Rnf20* genomic locus and the *Rnf20* targeting construct. **1c** PCR genotyping results showing distinct bands for wild-type (WT) and knockout (KO) alleles. **Fig. 6a** Schematic representation of the strategy used to generate *RNF20*^{+/-} A549 cells and PCR genotyping results showing distinct bands for WT and *RNF20* KO alleles in control and *RNF20*^{+/-} A549 cells. **Supplementary Fig. 2a** PCR genotyping results showing distinct bands for WT and *Rnf20* KO alleles in control and *Rnf20*^{+/-} MLE12 cells. **Supplementary Fig. 5a** Schematic representation of the strategy used to generate *Rnf20*^{+/-} LLC1 cells, and PCR genotyping results showing distinct bands for WT and *Rnf20* KO alleles.

Furthermore, we have incorporated RNF20 overexpression experiments to demonstrate the rescue of the observed phenotype, as follows: To determine whether the observed effects are a functional consequence of RNF20 loss, we next overexpressed RNF20. This overexpression reduced the elevated colony formation and migration abilities of *Rnf20*^{+/-} cells to levels comparable to those of control cells (Supplementary Fig. 2g, 2h). Moreover, RNF20 overexpression was sufficient to reduce HIF1 α and γ H2AX levels in *Rnf20*^{+/-} MLE12 cells to control levels (Fig. 5g, Supplementary Fig. 4e).

Supplementary Fig. 2

Supplementary Fig. 4

Supplementary Fig. 2g Clonogenic assay performed with control, *Rnf20*^{+/-} and RNF20 overexpressing (OE) *Rnf20*^{+/-} MLE12 cells. Representative images are shown on the left, quantification of the number of colonies is shown on the right (n=3). **h** Boyden chamber migration assay control, *Rnf20*^{+/-} and RNF20 overexpressing (OE) *Rnf20*^{+/-} MLE12 cells (left) and quantification of the number of migrated cells per field (right) (n=5). **Supplementary Fig. 4e** Western blot analysis of γH2Ax, HIF1α and RNF20 in total cell lysates from control, *Rnf20*^{+/-} and RNF20 OE *Rnf20*^{+/-} MLE12 cells.

-It is unclear why hypoxia does not result in a more substantial increase in HIF-1 protein levels in the MLE12 cells (Fig 4b). PHD or VHL inhibition may help show that this pathway is fully functional in the parental cell type.

Response: We are sorry for this confusion. We have now included a Western blot analysis of HIF1α in control and *Rnf20*^{+/-} MLE12 cells under normoxic and hypoxic conditions, shown together in Fig. 4b. The loss of a single *Rnf20* allele results in an increase in HIF1α levels

comparable to those observed in MLE12 cells cultured under hypoxia. Notably, *Rnf20*^{+/-} MLE12 cells exhibit even higher HIF1α protein levels under hypoxic conditions.

Reviewer #4 (Remarks to the Author):

Liu, Dobрева and colleagues here present their analysis of Rnf20 E3 ubiquitin ligase in lung cancer.

They show that Rnf20 heterozygote mice show markedly increased lung cancer incidence (adenocarcinoma and small cell). Rnf20 heterozygosity impaired DNA damage response signalling and resolution in CRISPR-targeted cell lines and in lung samples from the mice. This heterozygosity also led to phenotypic changes consistent with epithelial-mesenchymal transition (EMT), along with altered expression of key EMT-related genes. Among the major pathways impacted by Rnf20 heterozygosity were those controlled by the HIF-1 alpha transcription factor. The authors demonstrate greatly increased Hif-1 alpha levels in Rnf20 heterozygous cells, along with increases in glycolytic enzymes and activities. They then show that major phenotypes of the Rnf20 heterozygote cells can be suppressed by siRNA knockdown of Hif1-alpha. Seeking a mechanism for the role of Hif1-alpha in determining the outcome of Rnf20 heterozygosity, the authors found altered transcriptional activity of RNA Pol II, dependent on Rnf20-Hif-1 alpha, along with alterations in histone H2B monoubiquitination that indicated another route by which Rnf20 heterozygosity impacted gene regulation. The study concludes with the demonstration of a negative correlation between Hif1-alpha target expression levels and RNF20 in lung cancer patients, linking the paper's main findings with clinical outcomes.

This is a credible study that should be of interest to a broad readership. It has potential clinical ramifications, as well as opening new avenues for further investigation of the biology of Rnf20. My comments pertain mainly to technical controls and clarifications.

Response: We thank the reviewer for their appreciation of our study and for the constructive comments.

Major comments

1. Supplementary Fig. 1 should include formal ('genomic') demonstration of the successful targeting of the Rnf20 locus. Similarly, data confirming the expected disruption of the Rnf20 locus by CRISPR should be presented.

Response: We thank the reviewer for highlighting these deficiencies. We have now provided formal evidence of successful targeting of the *Rnf20* locus in Supplementary Fig. 1c. Similarly, PCR analysis confirming the expected deletion of the *RNF20/Rnf20* locus by CRISPR is

presented in Fig. 6a (A549), Supplementary Fig. 2a (MLE12 cells), and Supplementary Fig. 5a (LLC1 cells).

Supplementary Fig. 1a Schematic diagram of the *Rnf20* genomic locus and the *Rnf20* targeting construct. **1c** PCR genotyping results showing distinct bands for wild-type (WT) and knockout (KO) alleles. **Fig. 6a** Schematic representation of the strategy used to generate *RNF20*^{+/-} A549 cells and PCR genotyping results showing distinct bands for WT and *RNF20* KO alleles in control and *RNF20*^{+/-} A549 cells. **Supplementary Fig. 2a** PCR genotyping results showing distinct bands for WT and *Rnf20* KO alleles in control and *Rnf20*^{+/-} MLE12 cells. **Supplementary Fig. 5a** Schematic representation of the strategy used to generate *Rnf20*^{+/-} LLC1 cells, and PCR genotyping results showing distinct bands for WT and *Rnf20* KO alleles.

2. Details of the generation of the LLC1 and H82 *Rnf20*/*RNF20* heterozygotes should be provided (methodological information and controls for the disruption). This may not be needed for H82 cells if the description for Fig. 2e is incorrect.

Response: Details of the generation of *Rnf20*^{+/-} LLC1 cells are provided in Supplementary Fig. 5a and the *Methods* section. Indeed, the experiments performed with H82 cells were conducted following siRNA-mediated *Rnf20* silencing.

3. Rescue experiments should be presented for the MLE12 CRISPR cell line results. Performing these controls for the DNA damage response, proliferation and Hif-1 stabilisation experiments would exclude the possibility of an off-target impact or a clonal effect.

Response: We thank the reviewer for this comment. We have incorporated *RNF20* overexpression experiments to demonstrate the rescue of the observed phenotype, as follows: To determine whether the observed effects are a functional consequence of *RNF20* loss, we next overexpressed *RNF20*. This overexpression reduced the elevated colony formation and migration abilities of *Rnf20*^{+/-} cells to levels comparable to those of control cells (Supplementary Fig. 2g, 2h). Moreover, *RNF20* overexpression was sufficient to reduce HIF1 α and γ H2AX levels in *Rnf20*^{+/-} MLE12 cells to control levels (Fig. 5g, Supplementary Fig. 4e).

In addition, we have clarified in the *Quantification and Statistical Analysis* section that the sample sizes (n) presented in the plots represent individual clones.

Supplementary Fig. 2

Supplementary Fig. 4

Supplementary Fig. 2g Clonogenic assay performed with control, *Rnf20*^{+/-} and RNF20 overexpressing (OE) *Rnf20*^{+/-} MLE12 cells. Representative images are shown on the left, quantification of the number of colonies is shown on the right (n=3). **h** Boyden chamber migration assay control, *Rnf20*^{+/-} and RNF20 overexpressing (OE) *Rnf20*^{+/-} MLE12 cells (left) and quantification of the number of migrated cells per field (right) (n=5). **Supplementary Fig. 4e** Western blot analysis of γH2Ax, HIF1α and RNF20 in total cell lysates from control, *Rnf20*^{+/-} and RNF20 OE *Rnf20*^{+/-} MLE12 cells.

4. A rescue control experiment should be presented for the *Hif-1* RNAi cell line data.

Response: We have incorporated HIF1α overexpression experiments in *Rnf20*^{+/-}-shHif1a MLE12 cells, showing that similar to RNF20 loss, overexpression of constitutively active (non-degradable) human HIF1α overexpression construct (stbHIF1α OE) led to enhanced clonogenic growth, and elevated expression of its targets, *Slc2a1* and *Ldha* (Supplementary Fig. 4f-h).

Supplementary Fig. 4f Western blot analysis of HIF1α in total cell lysates from *Rnf20*^{+/-} MLE12 cells, or *Rnf20*^{+/-} cells expressing shRNA against *Hif1a* either alone or together with a constitutively active (non-degradable) human HIF1α overexpression construct (stbHIF1α OE). **g** Clonogenic assay performed with *Rnf20*^{+/-} MLE12 cells, or *Rnf20*^{+/-} cells expressing shRNA against *Hif1a* alone or together with varying amounts of stbHIF1α OE construct. Representative images are shown on the left, quantification of the number of colonies is shown on the right (n=3). **h** qPCR analysis of glycolysis and

hypoxia-related genes in *Rnf20*^{+/-} MLE12 cells, or *Rnf20*^{+/-} cells expressing shRNA against *Hif1a*, either alone or in combination with varying amounts of stblHIF1 α OE construct.

5. Is Hif-1 alpha monoubiquitinated? This should be tested as a potential direct link to *Rnf20* that might indicate a mechanism for the high Hif-1 levels seen.

Response: We have been investigating the hypothesis that RNF20 could function as an E3 ligase for Hif1 α ; however, we did not observe direct ubiquitination by RNF20. While we cannot completely rule out this possibility, our new data suggest that the decrease in RBX1 plays a critical role in the elevated HIF1 α levels. RBX1 is a component of the VHL tumor suppressor complex, which ubiquitinates HIF α subunits (HIF1 α , HIF2 α , and HIF3 α) and targets them for degradation under normoxic conditions^{14, 18}. Interestingly, *Rbx1* was specifically downregulated upon RNF20 LOF, but not following RNF40 depletion, which did not alter Hif1 α levels (Supplementary Fig. 6a).

To investigate whether decreased RBX1 levels are responsible for the elevated HIF1 α levels observed upon *Rnf20* loss, we overexpressed RBX1 in *Rnf20*^{+/-} cells. RBX1 overexpression led to a significant reduction in HIF1 α levels (Fig. 8d), reduced expression of HIF1 α target genes involved in glycolysis (Fig. 8e), and decreased clonogenic growth and migration capacity of *Rnf20*^{+/-} cells (Fig. 8f, 8g). Taken together, these data suggest that the reduction in RBX1 levels caused by *Rnf20* LOF is responsible for the increased HIF1 α levels and activity under normoxic conditions.

Fig. 8d Western blot analysis of HIF1 α in total cell lysates from *Rnf20*^{+/-} MLE12 cells or *Rnf20*^{+/-} stably expressing RBX1 (OE1 and OE2 stand for two stable pools). **e** Relative mRNA expression of *Rbx1*, *Slc2a1*, *Eno1* and *Ldha* in *Rnf20*^{+/-} MLE12 cells or *Rnf20*^{+/-} stably expressing RBX1. **f** Clonogenic assay with *Rnf20*^{+/-} MLE12 cells or *Rnf20*^{+/-} stably expressing RBX1 (left panels) and quantification of the number of colonies (right panel, n=3). **g** Boyden chamber migration assay with *Rnf20*^{+/-} MLE12

cells or *Rnf20*^{+/-} stably expressing RBX1 (left panels) and quantification of the number of migrated cells per field (right panel, n=5).

6. Given the complexity of *Rnf20*'s roles in different tissues, more information on the current understanding of the roles of *Rnf20* in cancer would be useful, either in the Introduction or in the Discussion. There are a number of recent papers that could be considered in the light of the results presented here, e.g. Duan et al. Nat. Commun. 2016 PMID: 27557628; Wegwitz et al. Cell Death Dis 2020 PMID: 33070155; Wang et al. Front Oncol. 2020 PMID: 33364200.

Response: We thank the reviewer for pointing out these studies, which are of great importance for our new data on the distinct functions of RNF20 and RNF40 in lung cancer. We have now discussed these studies in the discussion (page 19, lines 17-27) in the context of experimental evidence suggesting that, at least in lung cancer, RNF20 has a distinct role compared to RNF40.

Fig. 1f, h Normalized expression of *RNF20* (f) or *RNF40* (h) in normal lung and AD tissues in TCGA datasets. FPKM stands for fragments per kilobase of exon per million mapped fragments. **g, i** Overall survival (Kaplan-Meier plot) of lung AD patients⁶ expressing high vs low levels of *RNF20* (probe 222683_at) (g) or *RNF40* (probe 206845_s_at) (i). **Fig. 5 m** Western blot analysis of RNF20, RNF40

and H2Bub1 in normoxic conditions, as well as HIF1 α under normoxic and hypoxic conditions, in total cell lysates from control or *Rnf40* knockdown MLE12 cells. **n** Quantification of the number of migrated cells per field in a Boyden chamber migration assay with control and *Rnf40* knockdown MLE12 cells. **o** Clonogenic assay with control and *Rnf40* knockdown MLE12 cells. **p** qPCR analysis of genes involved in EMT (left) or glycolysis (right) in control and *Rnf40* knockdown MLE12 cells. **Supplementary Fig. 4k** Representative images of Boyden chamber-based migration assay control or *Rnf40*-shRNA-mediated knockdown MLE12 cells. Scale bars, 100 μ m. **l** Representative images of clonogenic assay performed with control or *Rnf40*-shRNA-mediated knockdown MLE12 cells.

RNF20 or RNF40 form a heterodimer that acts as the major E3 ligase responsible for histone H2Bub1 in mammalian cells⁵. Interestingly, in contrast to *RNF20* mRNA levels (Fig. 1f, 1g), *RNF40* mRNA levels were increased in AD patients (Fig. 1h) and higher *RNF40* levels significantly correlated with poor survival in the KMplotter lung adenocarcinoma dataset⁶ (Fig. 1i), suggesting distinct function of these two proteins in lung cancer. *Rnf40* silencing, in contrast to RNF20 LOF, did not affect HIF1 α levels (Fig. 5m), cell migration or clonogenic growth (Fig. 5n, 5o, Supplementary Fig. 4k, 4l), but significantly reduced the expression of some HIF1 α targets, such as *Fn1*, *Snai2*, *Vegfa* and *Eno1* (Fig. 5p). These findings indicate that RNF20 operates independently of RNF40 in suppressing cell growth, migration, and the regulation of HIF1 α -responsive genes, at least in lung epithelial and lung cancer cells. This contrasts with the roles of these proteins in breast cancer cells, where both RNF20 and RNF40 promote carcinogenesis^{19, 20, 21}. These differences may be due to their interactions with cell type-specific binding partners.

Minor comments

7. The legend for Supplementary Fig. 2e indicates that the H82 experiment uses siRNA to deplete RNF20, but the figure suggests that these are a heterozygote cell line. This should be clarified.

Response: Indeed, the experiments performed with H82 cells were conducted following siRNA-mediated *Rnf20* silencing. This is now clearly indicated in Fig. 6k-m (revised manuscript), and the legend of the figure.

8. The expected Mendelian number of null *Rnf20* mice from the heterozygote crosses performed should be provided, in support of the finding that no *Rnf20*(-/-) offspring were obtained.

Response: We have now included a table in Supplementary Fig. 1b that presents the percentage of expected genotypes from heterozygous crosses alongside the genotyped offspring.

9. All immunoblots should include size markers.

Response: We have included molecular size markers in all figure panels presenting Western blot analyses.

10. Statistical outcome for Supplementary Fig. 1d should be indicated.

Response: Statistical analysis has now been included in Supplementary Fig. 1f (corresponding to Supplementary Fig. 1d in the submitted manuscript). While a reduction in weight is observed, it does not reach statistical significance.

11. The legend to Fig. 3h should specify that these are the top 10 GO terms (if this is the case), i.e. that these are not selected from the Metascape analysis.

Response: Yes, Fig. 3h presents the top 10 GO terms. This information is now specified in the figure legend.

12. It should be clarified what the 2 lanes shown for each genotype in Fig. 4b represent (replicates, separate subclones, etc.?).

Response: We apologize for not clearly describing the experimental setup. The data presented in Fig. 4b represent analyses of individual clones. Similarly, throughout the manuscript, the data shown in the plots represent data obtained from individual clones. This information is now explicitly stated in the *Quantification and Statistical Analysis* of the *Methods* section.

13. It should be clarified what the gene expression changes are measured against in the heat map shown in Fig. 4d; what are the 3 lanes showing?

Response: The heatmap represents RNA-Seq analysis of control and *Rnf20*^{+/-} MLE12 cells (n=3, individual clones). This information is now specified in both the figure and its legend.

14. *Technical/ methodological details should be provided of the LC-MS experiments shown in Fig. 4d.*

Response: We apologize for the shortcomings in the *Methods* section. The technological details have now been added under the section *LC-MS/MS Data Acquisition and Analysis*.

15. *The immunoblot shown in Fig. 5g should be of better quality.*

Response: We have replaced the Western blot analysis in Fig. 5g with a revised blot that ensures more equal loading.

16. *'RRPM' should be explained in the legend to Fig. 6.*

Response: We apologize for the typographical error. RRPM should have been RPM, which stands for Reads Per Million mapped reads. This correction has now been indicated in the figure legend of Fig. 8c (Fig. 6l in the initial version of the manuscript).

17. *The curves should be placed in front of the individual datapoints in the graphs shown in Fig. 7a.*

Response: We have now added the Pearson correlation line in front of the individual data points (Supplementary Fig. 7b in the revised manuscript).

References:

1. Andrysik Z, Bender H, Galbraith MD, Espinosa JM. Multi-omics analysis reveals contextual tumor suppressive and oncogenic gene modules within the acute hypoxic response. *Nat Commun* **12**, 1375 (2021).
2. Gengenbacher N, Singhal M, Augustin HG. Preclinical mouse solid tumour models: status quo, challenges and perspectives. *Nat Rev Cancer* **17**, 751-765 (2017).
3. Kim J, *et al.* RAD6-Mediated transcription-coupled H2B ubiquitylation directly stimulates H3K4 methylation in human cells. *Cell* **137**, 459-471 (2009).
4. Ortmann BM, *et al.* The HIF complex recruits the histone methyltransferase SET1B to activate specific hypoxia-inducible genes. *Nat Genet* **53**, 1022-1035 (2021).
5. Zhu B, *et al.* Monoubiquitination of human histone H2B: the factors involved and their roles in HOX gene regulation. *Mol Cell* **20**, 601-611 (2005).
6. Gyorffy B, Surowiak P, Budczies J, Lanczky A. Online survival analysis software to assess the prognostic value of biomarkers using transcriptomic data in non-small-cell lung cancer. *PLoS One* **8**, e82241 (2013).
7. Nicholson AG, *et al.* The 2021 WHO Classification of Lung Tumors: Impact of Advances Since 2015. *J Thorac Oncol* **17**, 362-387 (2022).
8. Ferone G, Lee MC, Sage J, Berns A. Cells of origin of lung cancers: lessons from mouse studies. *Genes Dev* **34**, 1017-1032 (2020).
9. Meuwissen R, Linn SC, Linnoila RI, Zevenhoven J, Mooi WJ, Berns A. Induction of small cell lung cancer by somatic inactivation of both Trp53 and Rb1 in a conditional mouse model. *Cancer Cell* **4**, 181-189 (2003).
10. Williams BO, Remington L, Albert DM, Mukai S, Bronson RT, Jacks T. Cooperative tumorigenic effects of germline mutations in Rb and p53. *Nat Genet* **7**, 480-484 (1994).
11. Chen HJ, *et al.* Generation of pulmonary neuroendocrine cells and SCLC-like tumors from human embryonic stem cells. *J Exp Med* **216**, 674-687 (2019).
12. George J, *et al.* Comprehensive genomic profiles of small cell lung cancer. *Nature* **524**, 47-53 (2015).
13. Shema E, *et al.* The histone H2B-specific ubiquitin ligase RNF20/hBRE1 acts as a putative tumor suppressor through selective regulation of gene expression. *Genes Dev* **22**, 2664-2676 (2008).
14. Kamura T, *et al.* Rbx1, a component of the VHL tumor suppressor complex and SCF ubiquitin ligase. *Science* **284**, 657-661 (1999).

15. Pan J, Bishop T, Ratcliffe PJ, Yeger H, Cutz E. Hyperplasia and hypertrophy of pulmonary neuroepithelial bodies, presumed airway hypoxia sensors, in hypoxia-inducible factor prolyl hydroxylase-deficient mice. *Hypoxia (Auckl)* **4**, 69-80 (2016).
16. Tetik-Elsherbiny N, *et al.* RNF20-mediated transcriptional pausing and VEGFA splicing orchestrate vessel growth. *Nat Cardiovasc Res* **3**, 1199-1216 (2024).
17. Bray S, Musisi H, Bienz M. Bre1 is required for Notch signaling and histone modification. *Dev Cell* **8**, 279-286 (2005).
18. Gossage L, Eisen T, Maher ER. VHL, the story of a tumour suppressor gene. *Nat Rev Cancer* **15**, 55-64 (2015).
19. Duan Y, *et al.* Ubiquitin ligase RNF20/40 facilitates spindle assembly and promotes breast carcinogenesis through stabilizing motor protein Eg5. *Nat Commun* **7**, 12648 (2016).
20. Wang D, Wang Y, Wu X, Kong X, Li J, Dong C. RNF20 Is Critical for Snail-Mediated E-Cadherin Repression in Human Breast Cancer. *Front Oncol* **10**, 613470 (2020).
21. Wegwitz F, *et al.* The histone H2B ubiquitin ligase RNF40 is required for HER2-driven mammary tumorigenesis. *Cell Death Dis* **11**, 873 (2020).